# Reliable Ultrasonic Obstacle Recognition for Outdoor Blind Navigation

**Apostolos Meliones \*, Costas Filios and Jairo Llorente**

Department of Digital Systems, School of Information and Communication Technologies, University of Piraeus, 18534 Piraeus, Greece; kwstasfilios@gmail.com (C.F.); jairolozanollorente@gmail.com (J.L.)
\* Correspondence: meliones@unipi.gr; Tel.: +30-210-414-2762

**Abstract:** A reliable state-of-the-art obstacle detection algorithm is proposed for a mobile application that will analyze in real time the data received by an external sonar device and decide the need to audibly warn the blind person about near field obstacles. The proposed algorithm can equip an orientation and navigation device that allows the blind person to walk safely autonomously outdoors. The smartphone application and the microelectronic external device will serve as a wearable that will help the safe outdoor navigation and guidance of blind people. The external device will collect information using an ultrasonic sensor and a GPS module. Its main objective is to detect the existence of obstacles in the path of the user and to provide information, through oral instructions, about the distance at which it is located, its size and its potential motion and to advise how it could be avoided. Subsequently, the blind can feel more confident, detecting obstacles via hearing before sensing them with the walking cane, including hazardous obstacles that cannot be sensed at the ground level. Besides presenting the micro-servo-motor ultrasonic obstacle detection algorithm, the paper also presents the external microelectronic device integrating the sonar module, the impulse noise filtering implementation, the power budget of the sonar module and the system evaluation. The presented work is an integral part of a state-of-the-art outdoor blind navigation smartphone application implemented in the MANTO project.

**Keywords:** assistive application; outdoor blind navigation; blindness; vision impairment; obstacle detection; ultrasonic sensor; sonar; smartphone; Median filter; Kalman filter; MANTO project

## 1. Introduction

Over the last decades, there has been great development in the field of technology. Throughout the computer science history, Moore's law boosts constantly the microelectronic progress, increasing the processing power and performance of CPUs and SoCs (e.g., denser lithography, more and larger cache levels, RAM technology reducing refresh cycle and increasing operation frequency) and at the same time reducing the price of electronic products, allowing to create more powerful devices in smaller form factors. Some of the most striking advances that form a turning point in recent years are the Internet and the smartphone, assisting our modern life patterns. Thanks to these two advances, many applications and devices have been developed, which, in addition to providing leisure functions, provide great assistance in people's daily tasks. The smartphone has become a useful gadget and digital assistant to make purchases, manage jobs, entertain, learn and acquire knowledge, as well as exploit a lot of useful and innovative new applications. In fact, in many cases human beings depend significantly on them nowadays, simplifying and greatly enhancing their lives, including independent living for people with disabilities.

There is a large group of people who suffer some disability and the accomplishment of daily tasks is complicated for them. Within this group, people with visual impairment are guided only by their sense of hearing and touch. According to the World Health Organization, there are currently estimated to be 285 million people affected by visual

impairment, of which 39 million are blind. We are faced with the need to respond to these people and solve a problem that has been vastly restricting their lives: Insecurity on public roads [1]. The UN in 2006, in the Convention on the Rights of Persons with Disabilities (Article 24), manifested the need of a "universal accessibility", that is, that all citizens have the same opportunities, goods and services to carry out any type of activity, including the basic need to be able to go out on their own [2].

So far, the most common orientation and navigation tools that have been able to help blind people to get out safely are the service dog and the walking cane. However, the assistance provided by these two tools for safe mobility is limited and lacking reliable information. Multiple devices have been developed to help blind people navigate in outdoor and indoor places. The MANTO R&D project develops blind escort apps on smartphones which provide a self-guided outdoor and indoor navigation service with voice instructions [3,4]. The MANTO project significantly extends the BlindHelper pedestrian navigation system presented in [5], presenting the concept of vastly enhancing the user positioning accuracy and tracking for blind navigation, granting to this effort the best innovation paper award in ACM PETRA 2016. This work has been cited by several respectable extensive review papers in the fields of blind navigation [6,7]. The work presented in [3] includes also a rich literature review of blind indoor navigation systems.

The work in this paper presents in detail and evaluates an algorithm to analyze the data obtained by an ultrasonic sensor and decide the need to warn the blind person about near field obstacles. The proposed algorithm can equip an orientation and navigation device that allows the blind person to walk safely and be audibly informed of the near field obstacles. Such a device will help the blind feel more confident, as s/he will be able to detect obstacles via hearing before s/he can sense them with the walking cane, including obstacles that cannot be sensed at the ground level which can injure the blind or visually impaired (e.g., external air conditioning units, awnings, signs, windows, etc., at head level), and s/he will be warned of their size, position and the optimal way to avoid them. The current paper is an invited extended version of an ACM PETRA 2019 conference paper [8], which first presented the concept and a corresponding algorithm for motorized ultrasonic obstacle detection. The extended paper presents the implemented sonar obstacle detection system in the framework of a state-of-the-art outdoor blind navigation smartphone application developed in the MANTO R&D project [4]. Besides presenting the micro-servo-motor ultrasonic obstacle detection algorithm for the sake of clarity and completeness, it also presents the external microelectronic device integrating the sonar module, with a further focus on the impulse noise filtering implementation, the power budget of the sonar module and the system evaluation.

Several ultrasonic-sensor-based obstacle detection systems have been presented in the literature. The work presented in [9] describes a microprocessor-based system replacing the Braille keyboard with speech technology and introducing a joystick for direction selection and an ultrasonic sensor for obstacle detection. Another aged approach based on a microprocessor with synthetic speech output featuring an obstacle detection system using ultrasound is presented in [10]. The work presented in [11] highlights an autonomous navigational robot mounting a single board computer able to detect and avoid obstacles in its path using ultrasonic sensors. A sonar unit called the Smart Guide, which is a low-cost easy to use commercial product, has been developed by the Lighthouse for the Blind of Greece, a partner of the MANTO project [12], and is quite similar to the noncontact handheld tool for range sensing and environment discovery for the visually impaired in [13]. The basic argument is that a perception through exploratory movements (like those using a white cane) appears to be a natural procedure for environment discovery. The work in [14] combines several ultrasonic sensors attached to different body areas with adjustable sensitivity to reliably detect upcoming obstacles, including potholes and upward/downward stairs.

Ultrasonic-sensor-based obstacle perception has been proposed as well in indoor navigation applications. The work in [15] presents an indoor navigation wearable system

based on visual marker recognition and ultrasonic obstacle perception used as an audio assistance for the blind and visually impaired. The EU Horizon 2020 Sound of Vision project [16] implements a non-invasive hardware and software system to assist the blind and visually impaired by creating and conveying an auditory representation of the surrounding environment (indoor/outdoor) to a blind person continuously and in real time, without the need for predefined tags/sensors located in the surroundings. The main focus of the project is on the design and implementation of optimum algorithms for the generation of a 3D model of the environment and for rendering the model using spatial sound signals. The work in [17] presents a mobile wearable context-aware indoor map and navigation system with obstacle detection and avoidance for the blind and visually impaired using a depth sensor. The work presented in [18] developed a sensor module that can be handled like a flashlight by a blind user and can be used for searching within the 3D environment. Inquiries concerning object characteristics, position, orientation and navigation can be sent to a connected portable computer, or to a federation of data servers providing models of the environment. Finally, these inquiries are acoustically answered over a text-to-speech engine.

A comprehensive comparative survey among several ultrasonic and other sensor obstacle detection/avoidance systems, and the progress of such assistive technology for visually impaired people is included in [19]. The survey is based on various features and performance parameters of the systems that classify them into categories, giving qualitative and quantitative measures. In a similar way, reference [20] evaluates more recent research efforts using sensor-based walking assistants, including the use of ultrasonic sensors that provide surrounding information to visually impaired people through audio signal, vibration or both. These frameworks depend on the gathered information to recognize an obstacle and avoid it by calculating the distance between the users and obstacles using the velocity of the obstacles.

Besides such efforts exploiting ultrasonic sensors for the dynamic real-time detection of obstacles along the path of the visually impaired in outdoor and indoor environments, there are also other sensor technology approaches to address safe pedestrian navigation for the blind and visually impaired. These are mostly computer vision-based systems, e.g., [21–23], or infrared-enabled depth sensor-based systems, e.g., [24–26], executing on mainstream, computationally efficient mobile devices. Several such systems are extensively reviewed in [19,20,27].

Major motivations for the proposed approach against the limitations of the aforementioned and many other related work are: (1) The sophistication of the proposed method regarding the detection of object size and near field moving object tracking, and the subsequent short oral warnings for avoidance, and (2) that none of these efforts is an integral part of a holistic modern state-of-the-art reliable high precision wearable navigation system relying on a smartphone.

The structure of the paper is as follows. The background MANTO project is briefly presented in Section 2. The obstacle detection device of the outdoor blind navigation application developed in the MANTO project is presented in Section 3, with a detailed description of the obstacle detection algorithm following in Section 4. Section 5 provides a classification of typical obstacles that a blind person could come across along an outdoor pedestrian travel and explains how the proposed algorithm would react in each case. Section 6 discusses the implementation details on the smartphone and external embedded device using median and Kalman filtering algorithms for the removal of impulse noise in the sonar measurements and the associated performance issues, hardware footprints (CPU, memory, network, power consumption) and optimizations. Section 7 presents the evaluation of the proposed system and the outcome of validation trials. The paper ends with a "Conclusions and Discussion" section.

## 2. The MANTO Blind Navigation Project

The MANTO RTD project [4] aimed at the development and validation of two state-of-the-art navigation applications for the blind and visually impaired. The first application (Blind RouteVision) addresses the autonomous safe pedestrian outdoor navigation of the blind and visually impaired. Building upon very precise positioning, it aims to provide an accessibility, independent living, digital escort and safe walking aid for the blind. The application operates on smartphones, exploiting Google Maps to implement a voice-guided navigation service. The smartphone application is supported by an external embedded device integrating a microcontroller, a high precision GPS receiver and a servo-motorized sonar sensor for real-time recognition and by-passing of obstacles along the route of the blind person.

Additional application features include voice selected route destinations and supported functions, synchronization with traffic lights, exploitation of dynamic telematics information regarding public transportation timetables and bus stops for building composite routes which may include public transportation segments in addition to pedestrian segments, dialing and answering phone calls, emergency notification of carers about the current position of the blind person, weather information to help dress appropriately, etc. Furthermore, an innovative and challenging application component has been developed to provide real-time visual information to the blind person along the route exploiting machine- and deep-learning technology. The already developed application goes beyond the current state-of-the-art regarding blind navigation demonstrating unparalleled functionality and setting a precedent for its competitors, enhancing independent living of the blind in Greece and all over the world.

The second application (Blind MuseumTourer) enables blind and visually impaired persons to experience autonomous self-guided tours and navigation in museums and other complex public and private indoor spaces. Based on the outcome of the particular use case, the application will evolve long term to an indoor blind navigation system for complex public and private buildings, such as hospitals, shopping malls, universities, public service buildings, etc. The application executes on smartphones and tablets implementing a voice-instructed self-guided navigation service inside museum exhibition halls and ancillary spaces. It comprises an accurate indoor positioning system using the smartphone accelerometer and gyroscope and optional proximity sensors fit in the indoor space. The Blind MuseumTourer application has been deployed in the Tactual Museum of the Lighthouse for the Blind of Greece, one of the five tactual museums worldwide, in order to implement a brand new "best practice" regarding cultural voice-guided tours targeting blind and visually impaired visitors. A key objective of the project is to render museums completely accessible to the blind and visually impaired using the proposed technology and implement a success story which can eventually be sustainably replicated in all Greek museums and other large and complex public and private indoor spaces using accessible acoustic and tactual routes. It is noted that several public and private organisations have officially expressed their strong interest (via Letter of Intent) to evaluate the developed application.

The proposed applications aim to resolve accessibility problems of the blind and visually impaired during pedestrian transportation and navigation in outdoor and indoor spaces. Independent living has a key contribution to social and professional inclusion, education and cultural edification and quality of life of the blind and visually impaired. The project aims to provide an unparalleled aid to the blind all over the world so that they can walk outdoors safely and experience self-guided tours and navigation in museums and complex indoor spaces. In parallel, it also aims to enable and train cultural and other interested organizations to host and be accessible to people with such disabilities. The project outcomes will contribute decisively towards breaking social exclusion and address blind people of all ages.

The MANTO RTD project ("Innovative Blind Escort Applications for Autonomous Navigation Outdoor and in Museums") was funded by the national EPAnEK 2014–2020

Operational Programme Competitiveness-Entrepreneurship-Innovation under contract No. 593 and was implemented by the University of Piraeus Research Centre with the collaboration of the Lighthouse for the Blind of Greece and IRIDA Labs S.A. The project name originates from ancient Greek mythology: Manto was the daughter and blind escort of famous blind seer Tiresias.

## 3. Obstacle Detection Device

The outdoor blind navigation application developed in the MANTO project includes an external wireless microelectronic device for enhanced GPS positioning comprising a microcontroller, a Bluetooth module and a GPS module [4]. Figure 1 illustrates device versions 1 and 2. The older (black) device is built around the Atmega 2560 microcontroller and the U-blox NEO-6M GPS receiver. The new (white) version is built around the Atmega 328p microcontroller and the U-blox NEO-8M GPS module, demonstrating a reduced size, cost and power consumption, optimized memory management and battery level monitoring.

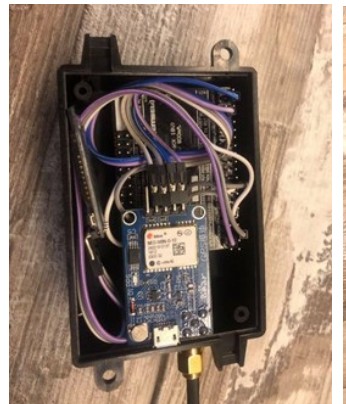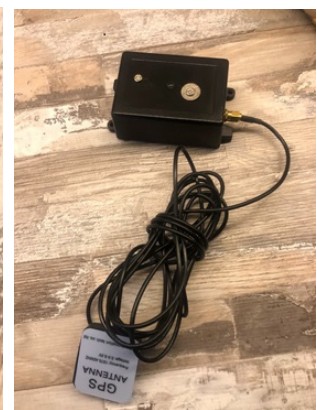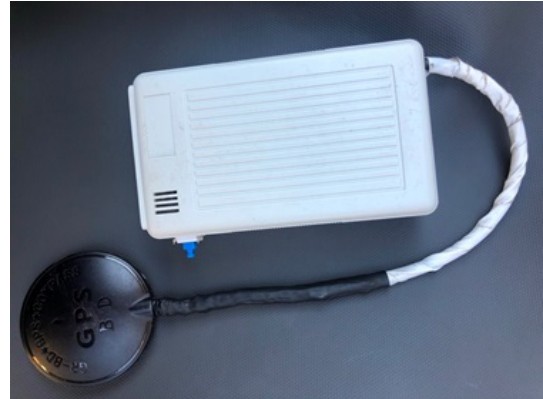

**Figure 1.** External wireless microelectronic device of the outdoor blind navigation application.

The external device can further interface a servo-motor mounting an ultrasound sensor comprising an obstacle detection module. The microcontroller will send the samples taken by the sensor and the GPS information via Bluetooth to the user's smartphone application. The smartphone application is responsible for making the calculations and estimations necessary to obtain the distance at which the obstacle is located, its size and where it is a moving object. With all this data, the application will decide whether to warn the user or not. In the ensuing, we briefly analyze the components, focusing mainly on the operation of the ultrasonic sensor. Figure 2 illustrates the prototype detection system that integrates the cost-efficient HC-SR04 ultrasonic sensor mounted on a MG90S micro-servo-motor.

The microcontroller is the cornerstone of the integrated application since it executes the code responsible for receiving the geographical coordinates of the person in motion, the sonar operation, including the receiving of the ultrasonic sensor detections and the continuous rotation of the servo-motor, as well as the sending of data from the device to the smartphone application via Bluetooth. The correct choice of the microcontroller is a definitive step towards the implementation of the system, with an impact on the total cost and future updates of the embedded application.

The Bluetooth module is interconnected with the microcontroller through a UART interface in 9600 baudrate, which promises to send the application data from the microcontroller to the smartphone application through serial communication. It is a basic component since, as mentioned, the obstacle detection information will be collected by the sonar device and the Bluetooth module will be responsible for sending it to the application.

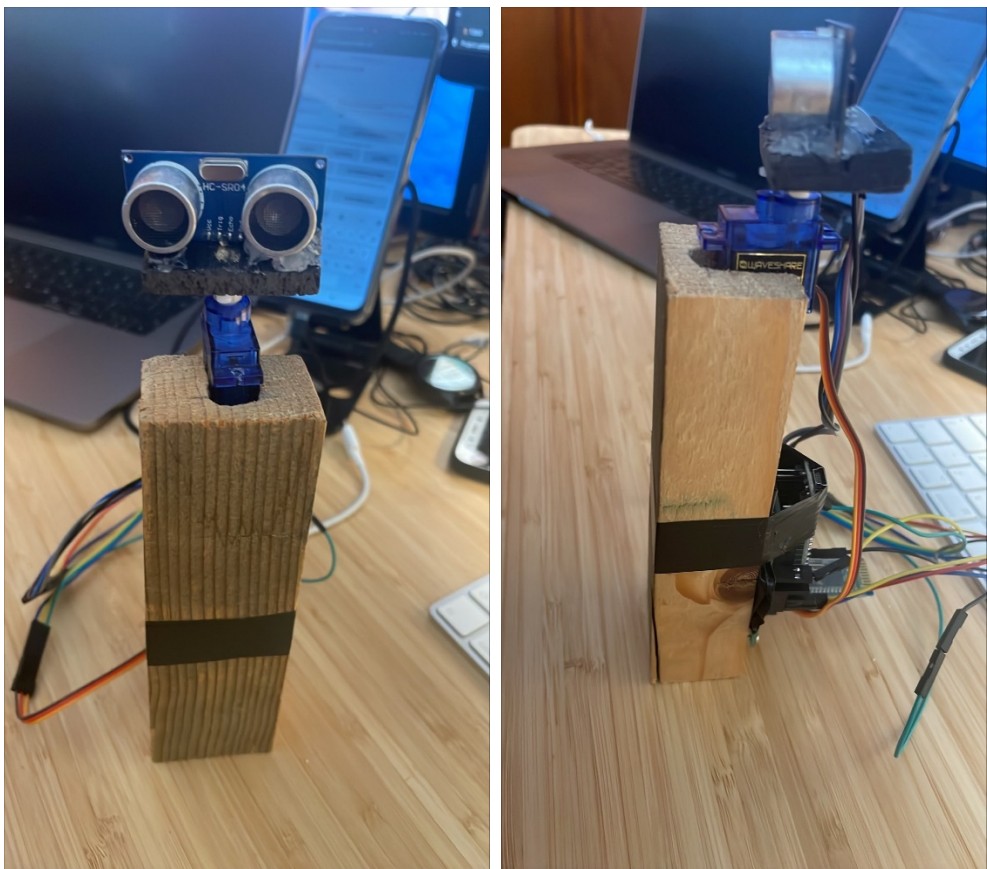

**Figure 2.** Prototype system for detailed obstacle detection.

The external device integrates a high precision GPS receiver, exploiting up to 16 geostationary satellites and achieving a centimeter-level location accuracy in the demanding context of pedestrian navigation for people with visual disabilities. The trials reported in [4,5] measured deviations, which are crucial for pedestrian mobility, of the order of 10 m between the locations reported by the smartphone integrated GPS tracker and the corresponding real geographic coordinates, while the deviations of the external GPS tracker were less than 0.4 m, receiving signal from 11 satellites. This module provides useful information to the smartphone outdoor blind navigation application about the user's location. The app is aware of the latitude, altitude, speed, bearing, date, time and number of satellites used.

*Ultrasonic Sensor*

The outdoor blind navigation application is using an ultrasonic ranging module, which is integrated in the external embedded device of the smartphone application, to determine the distance between the sensor and the closest object in its path. The sensor transmits a sound wave at a specific frequency. It then listens for that specific sound wave to bounce off at an object and come back. This sensor provides the pulse information required. That information is analyzed by the application to know the distance and the size of the object, to warn the blind user and to report a way to avoid it.

The selected HC-SR04 ultrasonic sensor (see Figure 3) provides a 2 cm–400 cm measurement function via an effective angle of 15 degrees, and its ranging accuracy can reach 3 mm. It has been selected for the development of the obstacle detection device mainly because of its low cost (less than $2), ease of use, sampling speed and its small weight and size, which make it a good choice for wearables. It has a single control pin responsible for transmitting the ultrasonic burst. This pin should be set high for 10 us, at which point the ultrasonic sensor will transmit an eight-cycle sonic burst at 40 kHZ (3.125 us). After

that, the Echo pulse output data pin, used in taking distance measurements, will be set to high. It will stay high until the ultrasonic burst is detected back, at which point it will be set to low.

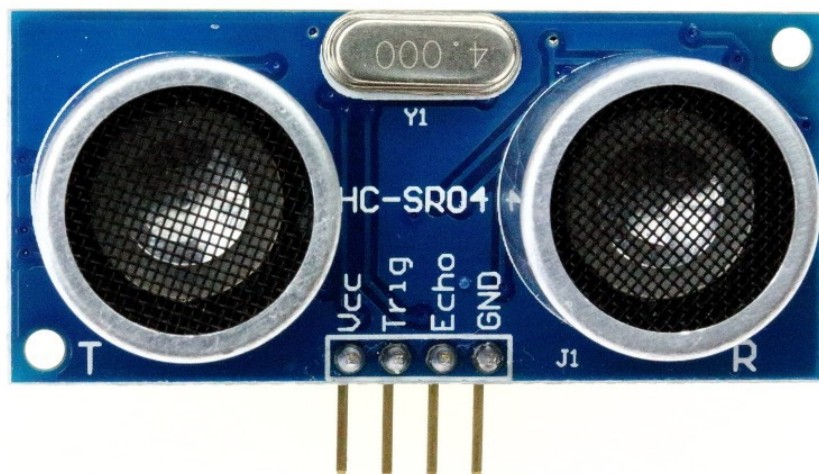

**Figure 3.** Ultrasonic sensor.

The distance to the object is calculated by keeping track of how long the Echo pin stays high. The time Echo stays high is the time the burst spent traveling. The time spent until the burst finds the object is half because the wave travels the same distance twice. As an example, consider that the Echo pin stays high for 12 ms. The distance between the ultrasonic sensor and the object is:

$$S = V * T * S = 343 \, \frac{m}{s} * 6 \, \text{ms} \; \rightarrow S = 2.058 \, \text{m} \tag{1}$$

Another case is when the sensor does not detect any object. The maximum time that the sensor can wait for the Echo pulse is 38 ms. After that time, the sensor determines there is not object in front. The ability of the sensor to detect an object depends on the object's orientation to the sensor. If an object does not present a flat surface to the sensor, then it is possible the sound wave will bounce off the object in a way that it does not return to the sensor.

## 4. Obstacle Detection Algorithm

Following the introduction of the different components of our obstacle detection device, this section explains the operation of the implemented obstacle detection algorithm. The angle of the sensor is increased to cover additional space. Therefore, the sensor is mounted on a servo-mechanism, which will allow the ultrasonic sensor to rotate in such a way that it covers 15 degrees to the left and 15 degrees to the right from the global position; that is, the view angle of the sonar will be 45 degrees (see Figure 4).

The sensor will take nF1 frontal samples, then it will take nL left samples, then it will take nR right samples and finally, the sensor will take nF2 frontal samples. All data (sensor data and map data) will be stored in an array and will be sent to the smartphone application via Bluetooth. Then the application will analyze the detection data and make the calculation to determine the necessary parameters. An example of a loop that the microprocessor will execute is the following:

1. Instructions for obtaining GPS data.
2. Instructions for obtaining the nF1 frontal samples.
3. Turn left the servo-motor.
4. Instructions for obtaining the nL left samples.
5. Turn right the servo-motor.
6. Instructions for obtaining the nR right samples.

7.  Turn the servo-motor to the initial position.
8.  Instructions for obtaining the nF2 frontal samples.
9.  Send the data to the application via Bluetooth.

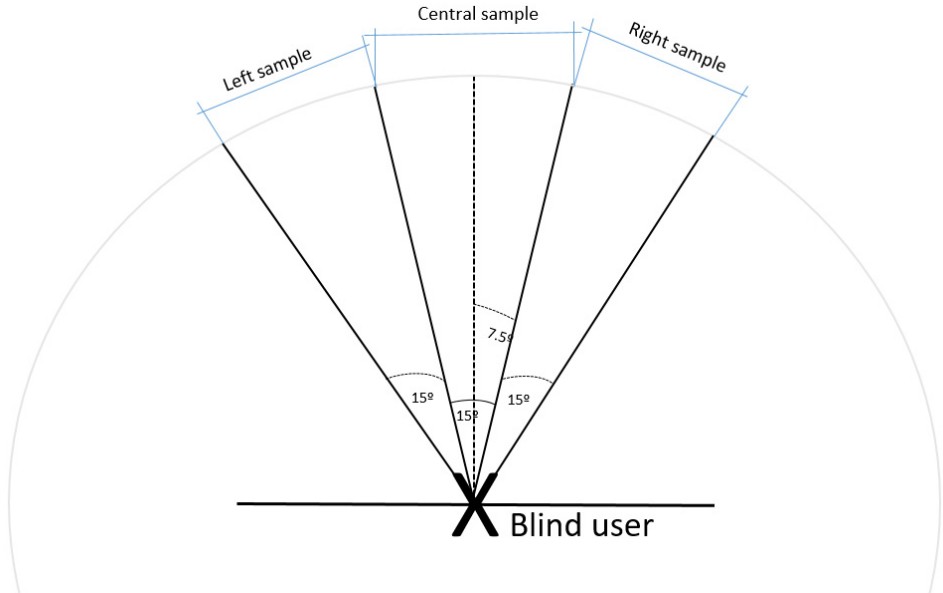

**Figure 4.** Angle detection of the servo-sonar system.

The application will create a new data object each time it receives data via Bluetooth. With that object, the pertinent calculations will be carried out and, if appropriate, the object will be stored in an arrayList. The objective is to have an arrayList with objects of the same obstacle. For example, after the first reception of data, it can be determined that the obstacle is 3 m away. The next data object received determines the object distance at 2 m, and the next one at 1 m. These three objects are stored in the same arrayList because they belong to the same obstacle. If the distance of the next data object received is bigger than the distance of the last element in the arrayList, it means that it is a new obstacle, and the previous data will be eliminated. Having the data of the same obstacle in an arrayList further helps determine whether the object is moving. In addition, consulting previous samples helps specify in a more exact way the width of the obstacle. These details will be analyzed in the ensuing.

Regarding the number of samples taken, it is a parameter that can be varied. All the samples will be in an array, and the first four elements of that array will indicate the amount of each type of sample that will be taken (front1, left side, right side and front2). An example with few samples of each type to help the understanding of the algorithm is depicted in Figure 5.

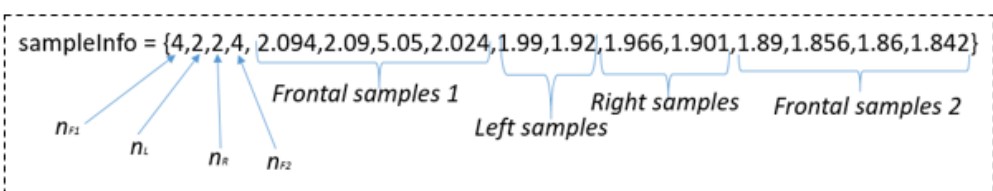

**Figure 5.** Sample data array structure.

The number of samples parameters (nF1, nL, nR, nF2) help the application dynamically determine the number of frontal and lateral samples that the microcontroller will order the sensor to take in each loop, allowing greater versatility in the amount of data that we need to obtain. This dynamic allocation will be analyzed later.

The proposed outdoor blind navigation obstacle detection algorithm is illustrated in Figure 6. The sonar data will be received by the application. This data is used to calculate the user's distance to the obstacle, the size of the obstacle and the speed of the user. Subsequently, the data received is compared with the previous data stored in the data array. In that data array, there is only data referring to the same obstacle. After the comparison, it is determined whether the new detected obstacle is the same as the obstacle stored in the array. Otherwise, the array data is deleted, and the new data received is added to the array. In case the algorithm determines that it belongs to the same obstacle, the received data is added to the array. Through comparison with the other data in the array, the movement of the obstacle is calculated, and in addition the new number of samples for the next burst is established. The algorithm then determines whether it is necessary to warn the user. The following paragraphs analyze how the calculations are made, starting with the distance calculation.

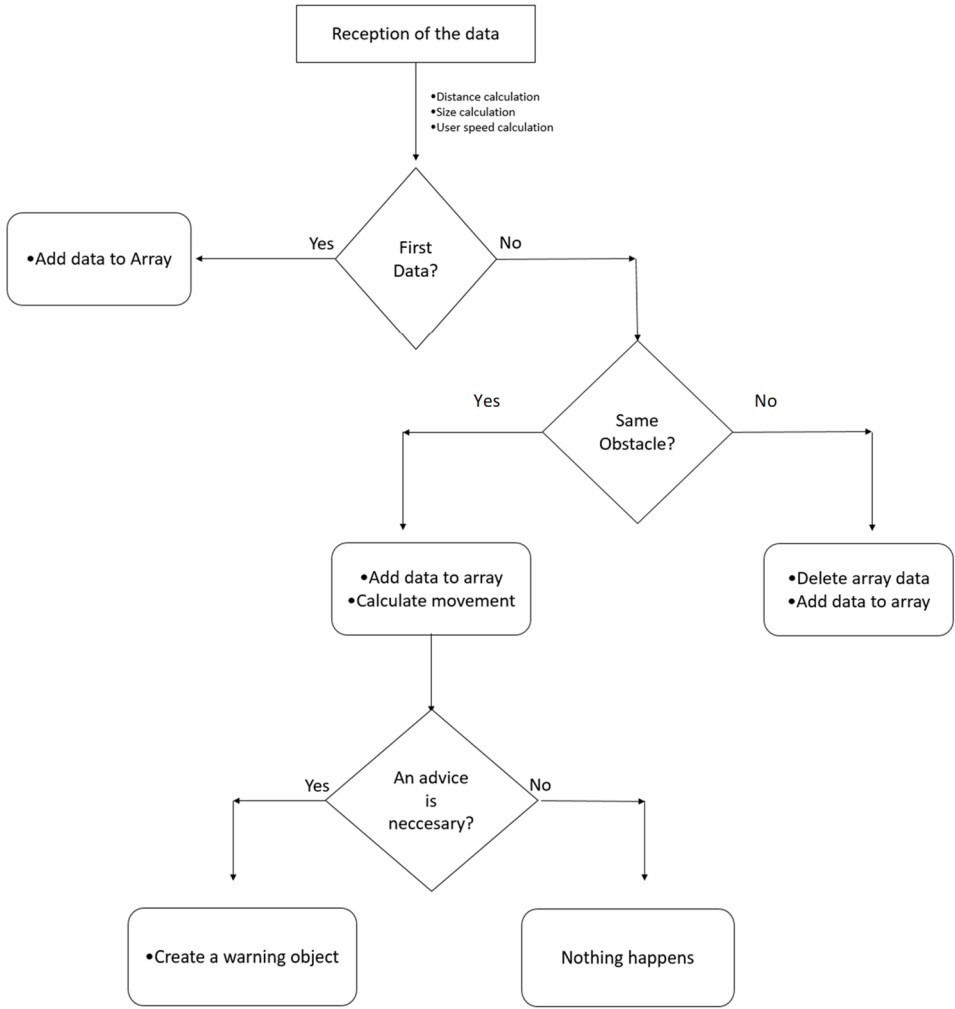

**Figure 6.** Flowchart of the proposed obstacle detection algorithm.

### 4.1. Obstacle Distance Calculation

The calculation of the distance is based on the correct interpretation of the front samples. The lateral samples are not used to calculate the distance the obstacle is located. These samples will be useful when calculating the size. The algorithm analyzes whether the next sample is in a close range comparing it with the previous samples and at the same time performs an averaging of the samples giving greater priority to the newest samples.

Let us consider the *sampleInfo* vector depicted in Figure 5. It has four frontal samples. The algorithm will detect the first two samples, compare their distances and if they are

similar, it will determine the distance of the object as the average between the two values. With the following samples, the algorithm will compare the distance determined previously with that of the new sample. If the new distance is within a range $-\varepsilon < 0 < +\varepsilon$, then the average will be made between the distance we had obtained and the new sample. Otherwise, it will be ignored. In this example, the third sample (position 6 of the array) is out of range (5.05 m); therefore, it would not be considered for the distance calculation. The fourth sample, on the contrary, would be considered in the *finalDistance* calculation.

Subsequently, the *finalDistance* determined by the first burst of front samples is averaged with the information of the second burst of front samples in a similar way. Higher priority will be allocated to the final samples (samples that occurred less time ago) than to the older ones. If the number of samples taken increases, the information will be more reliable; however, the microcontroller loop will take more time to capture the data and the transmission delay will increase. In addition, the delay of Bluetooth communication must be considered since the user during that period is also moving. Assuming a uniform user movement; therefore:

$$finalDistance = finalDistance - userSpeed * delayOfComm \tag{2}$$

The *userSpeed* and *delayOfComm* parameters are determined thanks to the GPS module and timestamp, respectively. In the example of Figure 4, the calculated *finalDistance* will be 1.86475 m, while for a user speed of 0.78 knots (approximately 0.4 m/s) and a communication delay of 20 ms, *finalDistance* = 1.85695 m.

### 4.2. Obstacle Width Calculation

This section describes the process of the obstacle angle detection and width calculation. The angle calculation is necessary to calculate the width of the obstacle.

#### 4.2.1. Obstacle Angle Detection

First, it is necessary to distinguish if an object has been detected in the left lateral samples and if an object has been detected in the right lateral samples. This is done by comparing the obtained measure with a range around the previously calculated *finalDistance*. If it is within the range, then the algorithm increases by one a reliable sample counter of the analyzed side. At the end, if the counter is greater than half the amount of the side samples, it is determined that there is an obstacle on that side. Subsequently, the *detectionOfSample* attribute will be assigned a value depending on whether the side samples have detected an obstacle:

1. Right and left side samples have not detected an obstacle.
2. Just the left samples have detected an obstacle.
3. Just the right samples have detected an obstacle.
4. Left and right samples have detected an obstacle.

Once this is known, it is possible to interpret the angles at which the sensor pointed it has detected an obstacle. In the next paragraph, the mathematical calculation of the width of an object according to its attributes *finalDistance* and *detectionOfSamples* will be explained.

#### 4.2.2. Obstacle Width Calculation

In order to explain the procedure, the simplest case is assumed, being the lateral samples not having detected any obstacle. In that case, *detectionOfSample* = 0 and *data.angle* = 15 degrees. What happens is that the exact position in which the object is found is not known because if the sensor covers an angle of 15 degrees, mathematically, there are infinite points at 1.86475 m (the example distance that we have already obtained) in which the obstacle can be found. We can see this in Figure 7.

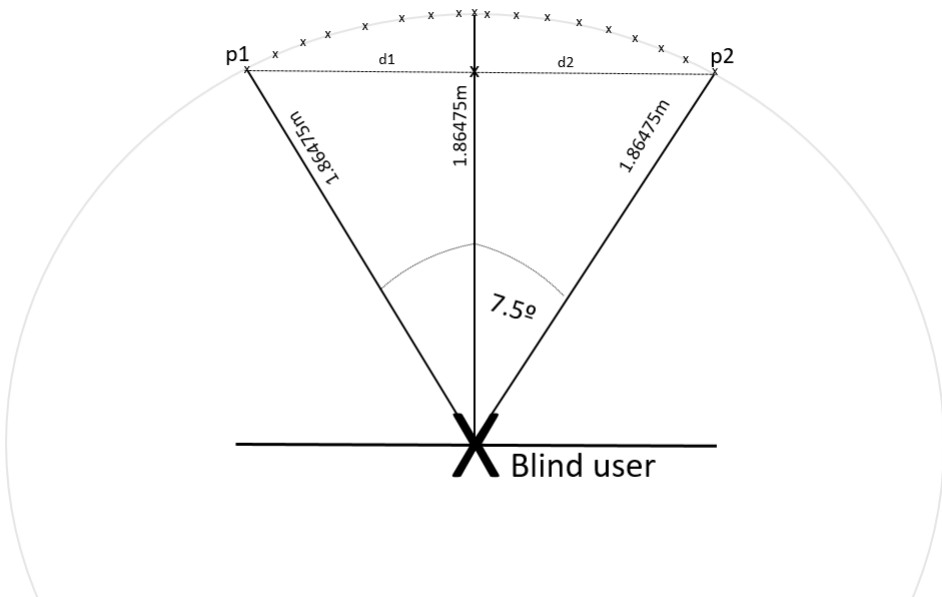

**Figure 7.** Different frontal positions of the obstacle.

Knowing that the arc is the one formed by an angle of 15 degrees in a circle of radius *r* = 1.86475 m, the distance between the leftmost point (p1) and the rightmost point (p2) can be easily calculated by applying some simple trigonometric formulas.

$$sin\left(\frac{a}{2}\right) = \frac{d_1}{r} \tag{3}$$

Being $\alpha$ = 15 degrees, *r* = 1.86475 m, $d_1 = d_2$
Substituting at Equation (3) yields $d_1$ = 0.2434 m.
Therefore, $\mathrm{d}p_1p_2$ = 0.48679 m.

With this data, for the time being it can be assumed that the object has a width of 0.48679 m, and that to the left and right of that distance there is no object, so the user could avoid that obstacle simply by moving that distance to the left or to the right. This will be discussed in the ensuing paragraphs.

The procedure if the lateral samples have detected an obstacle is similar. The only difference is that the angle that the sensor is covering ($\alpha$) is not 15 degrees. If the obstacle is detected in one lateral position, the angle is 30 degrees. If it is detected in the two lateral positions, the angle will be 45 degrees. Table 1 depicts possible widths for certain distances and for certain detections in lateral samples.

It can be observed that the estimated size which is calculated depends on the *finalDistance* previously calculated. Therefore, an object detected at 4 m with 3 angles will have a size of 3.061467 m. However, in the next burst, the same obstacle can be detected at 1 m, with a size of 0.765366 m. This problem is solved by storing the data of the same obstacle in the arrayList, and when the warning is elaborated, the largest size within the arrayList that is stored will be the one determined.

**Table 1.** Object size for different distances and angles.

| Distance, Radius(m): r | Angle(°): $\alpha$ | Object's Size (m): s |
|:---:|:---:|:---:|
| 1 | 15 (Just the front) | 0.261052 |
| 1 | 30 (front + 1 lateral) | 0.517638 |
| 1 | 45 (front + 2 lateral) | 0.765366 |
| 2 | 15 | 0.522104 |
| 2 | 30 | 1.035276 |
| 2 | 45 | 1.530733 |
| 3 | 15 | 0.783157 |
| 3 | 30 | 1.552914 |
| 3 | 45 | 2.2961 |
| 4 | 15 | 1.044209 |
| 4 | 30 | 2.070552 |
| 4 | 45 | 3.061467 |
| $2 \times \sin(a/2) = s/r$ | | |

An angle of 15° means the sensor has not detected anything in the lateral samples. An angle of 30° means the sensor has detected something only in the left or right lateral. An angle of 45° means the sensor has detected something in both laterals.

### 4.3. User Speed Calculation

The data that is received from the GPS module follows the NMEA protocol (acronym of National Marine Electronics Association), which are standard sentences for the reception of GPS data. One set of them and the most used is the $ GPRMC sentences, which have the following structure:

$GPRMC,044235.000,A,4322.0289,N,00824.5210,W,0.39,65.46,020615,,,A*44

This data frame contains the following variables:

- 044235.000 represents the time GMT (04:42:35).
- "A" is the indication that the position data is fixed and is correct. "V" would be invalid.
- 4322.0289 represents the longitude (43° 22.0289′).
- N represents the North.
- 00824.5210 represents the latitude (8° 24.5210′).
- W represents the West.
- 0.39 represents the speed in knots.
- 65.46 represents the orientation in degrees.
- 020615 represents the date (2 June 2015).

Therefore, the speed in knots reported by the GPS module can be obtained directly. The external embedded device sends this information to the smartphone application. Knowing that 1 knot is equal to 0.51444 m/s, the movement of the user can be calculated easily: 0.200633 m/s. This information is stored in the *userSpeed* attribute.

### 4.4. Adding the Object to the Calculation

Once the distance and the width of the obstacle have been calculated, all the necessary basic parameters are obtained, and it is possible to store the data in an array structure. As mentioned before, the idea is to store in an arrayList the objects belonging to the same obstacle. With the distance of the obstacle, it is possible to compare it with the distance of previously received data and determine if it is the same or a different obstacle.

What the algorithm will do first is to determine the size of the data array. If it is empty, it is because it is the first data reception; therefore, the new data object will simply be added to the arrayList. If it is not empty, it will obtain the final distance of the last data element of the array, i.e., the most recent data. If that distance is greater than the distance of the new element, it means that the obstacle is the same, so the data object will be added to the array. In addition, the function *SpeedObstacle* will be called. This function will calculate the

speed of the obstacle based on the previous data collected (see following paragraph). If the distance is not bigger, it means that it is a new obstacle. The algorithm will then delete the components of the array and set the new data received as the first element.

### 4.5. Obstacle Motion Calculation

Once it is known that the new obstacle detected is the same as the previous one, the kind of motion it performs can be detected. This information can be figured out because the user speed as well as the current and previous object distance are known. The time that elapses between each data reception is taken into account through a counter called *delayBetweenFirstSamples*. Therefore, it is possible to estimate the distance at which the obstacle should be found in the current sample. The algorithm estimates the position in which the obstacle should be found taking into account the data of the previous reception. There are three different cases:

1. The real distance is shorter than the expected distance of a static obstacle (minus a toleration range), which means that the obstacle is approaching. The variable *kindOfMovement* will be assigned the value "1".
2. The real distance is larger than the expected distance of a static obstacle (plus a toleration range). The obstacle is moving away, and it is faster than the user; therefore, there is no need to warn him. For this situation, the variable *kindOfMovement* will be assigned the value "2". In case the object moves away at a lower speed, it will be detected as a static obstacle. The difference will be that the smartphone application will collect many more data objects of that same obstacle.
3. Neither of the two previous assumptions is fulfilled, the expected distance is similar to the real one, which means that the object is static. The variable *kindOfMovement* will be assigned the value "0".

In cases 1 and 2 it is not necessary to warn the user since an obstacle that moves away at a higher speed is not a problem and an obstacle that is approaching in 99% of cases will likely be a human person who can see the blind person and move away. Case 3 requires warning the user. The determination of the warning to the user will be explained in the next paragraph.

### 4.6. Warning the User

In case 3 mentioned before the application must notify the user. However, it is not necessary to warn the user that s/he will run across an obstacle within 4 m since that obstacle will be detected again in the next iteration at a shorter distance. The algorithm sets the notification threshold at 3 m. If these assumptions are met, a new object of the warning class is instantiated. The algorithm will set the distance and size used by the instantiated warning class as the distance calculated in the last component of the arrayList, and as the largest size in the arrayList, respectively. To avoid the obstacle, the *detectionOfSample* variable is used, which indicates the angle at which an obstacle was detected, and through a switch instruction the cases are distinguished and the user is notified.

### 4.7. Setting the New Number of Samples

It has been said that the number of samples can be variable and that the application can set it. This is explained in the ensuing. An iterator traverses the arrayList. The *detectionOfSample* parameter indicates, as explained before, the position of the detected obstacle.

In case it has been detected that the obstacle is in the frontal position and, e.g., on the right, for the next sample the lateral samples on the right would not be needed since it is assumed that the obstacle will remain on the right. It is expected that the object will be closer and perhaps another obstacle appears to the left; therefore, the microcontroller will be instructed not to take samples in that direction.

In case it is detected that there is an object on both sides, the application will tell the microcontroller not to calculate lateral samples because only the frontal ones are of interest.

In case a new obstacle is detected, the sample number parameters are reset to the default parameters. This is done after deleting the data array.

## 5. Types of Obstacles

Having explained the details of the detection algorithm, this section distinguishes the different types of typical obstacles that a blind or visually impaired could come across along his route and explain how the algorithm would react. Finally, we present some types of obstacles that cannot be detected by the device.

### 5.1. Fixed Obstacles

These are static obstacles, which according to their exact position will require a warning to the user.

#### 5.1.1. Small Width Obstacles

Such obstacles fit within 15 degrees of sensor view, such as in the case of Figure 8a; therefore, the lateral sampling does not detect anything. Such objects can be easily detected, and the user can be warned accordingly. The sensor will detect the object through the frontal samples; however, the device will not detect anything with the lateral samples. The first object detection will be stored in an arrayList. The application will receive more object detections, and when the distance is less than 3 m, the user will be warned that he can avoid the obstacle by the left or by the right.

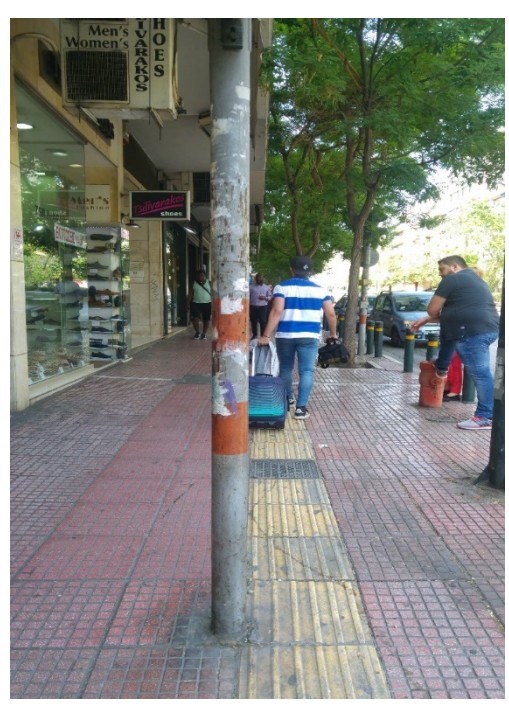

(**a**)

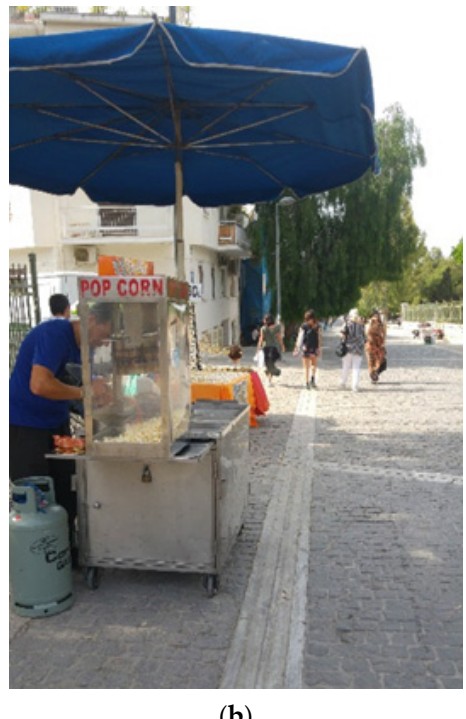

(**b**)

**Figure 8.** (**a**) Example of small width obstacle; (**b**) Example of medium width obstacle.

#### 5.1.2. Medium Width Obstacles

These obstacles are detected by the sensor in the central position and at least one of the lateral positions. In Figure 8b, the sensor will detect that there is something in front through the frontal samples. Furthermore, it will detect that there is an obstacle in the left-lateral samples and will not detect anything on the right. In that case, the application will decide that it is not required to take more left-lateral samples because the size of the obstacle will be calculated through the distance of the farthest object whose range cover a bigger angle. For example, if this object is detected at 4 m, and left-lateral samples have

detected an obstacle, the size will be 2.07 (according to Table 1). The next object detection will be received at a shorter distance, for example, 2 m, and the size calculated would be 1.04 (according to Table 1). Therefore, it is not useful to take the left-lateral samples once the previous left-lateral samples have been detected.

### 5.1.3. Large Width Obstacles

These obstacles are detected by the sensor in all positions, such as in the example cases in Figure 9. Both the front and side samples detect an obstacle. In that case, the indication reported to the user is that there is an obstacle of large width, which could be a wall. It is expected that this type of situation will be handled through another type of technology, such as through consulting the GPS and map info or by photographic interpretation of the outdoor, before, eventually, the blind or visually impaired person senses in close proximity the large object through the cane.

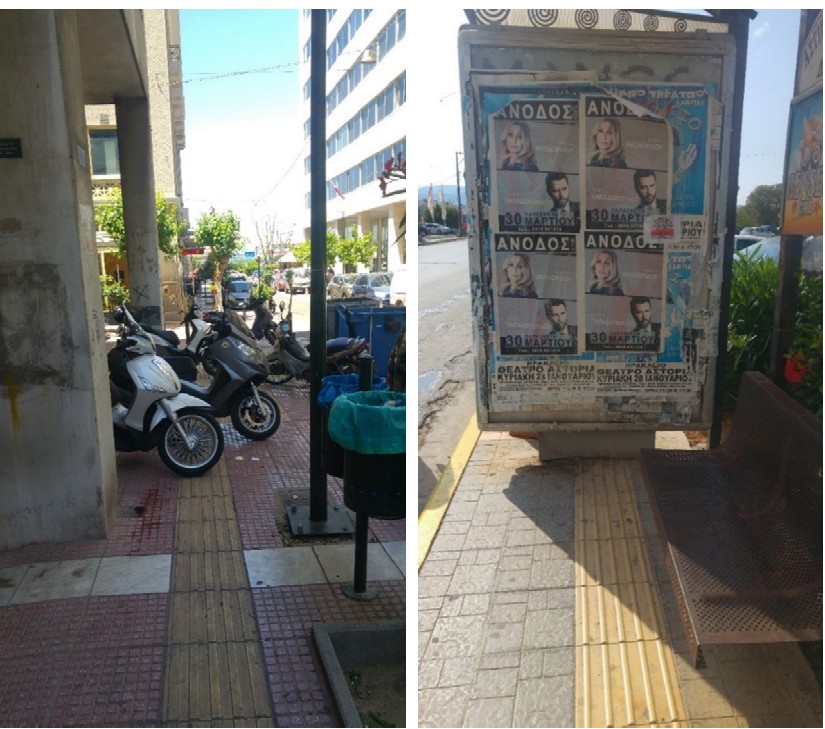

**Figure 9.** Examples of large width obstacles.

### *5.2. Not Fixed Obstacles*

These are moving objects that can be distinguished depending on their trajectory.

### 5.2.1. Movement in the Same Direction

1.  *Faster than the user*: It can be a person who is walking in the same direction as the user but at a higher speed. These kinds of obstacles are not important because there is not a risk of collision. Therefore, a warning will not be sent by the application to the user.
2.  *Slower than the user*: Such an obstacle will move at a fairly slow speed and will be treated as a static obstacle. A warning will be sent by the application to the user.

### 5.2.2. Movement in the Opposite Direction

These obstacles are directed towards the user; 99% of cases will likely be a human coming close. In this case, the application is not warning the user, as the system considers that the moving obstacle is able to detect the presence of the user and avoid him/her.

*5.3. Special Cases*

This section describes obstacles of the following types that are not as easy to detect or it is not as easy to calculate the size or the correct way to avoid them.

5.3.1. Not Detected Obstacles

In this frame, we can find obstacles that unfortunately will not be detected by the ultrasonic sensor, either because they do not have an adequate height or because their width is very small. In the case of Figure 10a, the sensor will never be able to detect that there are stairs in front because they are too low. Similarly, in the case of case of Figure 10b, the sensor will send ultrasonic pulses that will not find an obstacle from which to bounce and receive the pulse back; therefore, it will determine that there are no obstacles. In both cases the sensor samples will never detect any obstacle; therefore, the user will not be warned. Such obstacles at the ground level will be eventually detected safely by the trained blind person using the white cane. Another obstacle that probably cannot be detected by the sonar is when, in the case of Figure 8a, the post width is very small, as, e.g., the pole of a signpost (see Figure 9a). In that case, the sensor will send pulses which will not bounce correctly in the signpost because its width is too small. This can lead to the mistaken determination that there is no object in front.

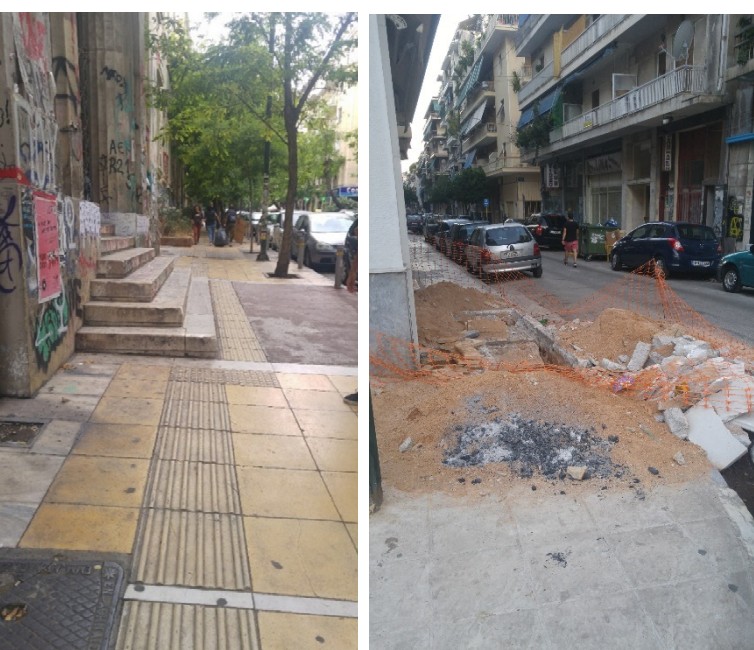

**Figure 10.** Not detected obstacles.

A possible solution to the problem of detecting obstacles and hazards at the ground level using ultrasonic sensors is discussed in [14], a work that combines several ultrasonic sensors attached to different body areas with adjustable sensitivity to reliably detect upcoming obstacles. It is claimed in this work that potholes and upward/downward stairs can be detected using sensors adjusted at the lower part of the leg. Another system that claims it can detect such obstacles at the ground level is the NavGuide system [28], which integrates several cost efficient ultrasonic sensors at the shoes of the blind person. This system can detect ascending staircases, as well as a wet floor when the blind first steps on it.

5.3.2. Obstacles That Present Some Difficulty to Analyze

In some cases, either due to the shape of the obstacle itself or due to the street surrounding, it is impossible to advise the user in a correct way. In the case of Figure 11a, we

observe that the sensor will detect that there are obstacles in any of the angles, and it will not be able to find an escape route due to how narrow the road between the car and the wall is. In the case of Figure 11b, the sensor will first detect that there is an obstacle in front and to the right and warn the user that to avoid it s/he must go to the left. However, it will end up detecting that there is an object also on the left, and it cannot find a reliable path for the user. The case of large width objects, such as in Figure 9, cannot be handled effectively as well, as the application cannot decide a reliable way for the user.

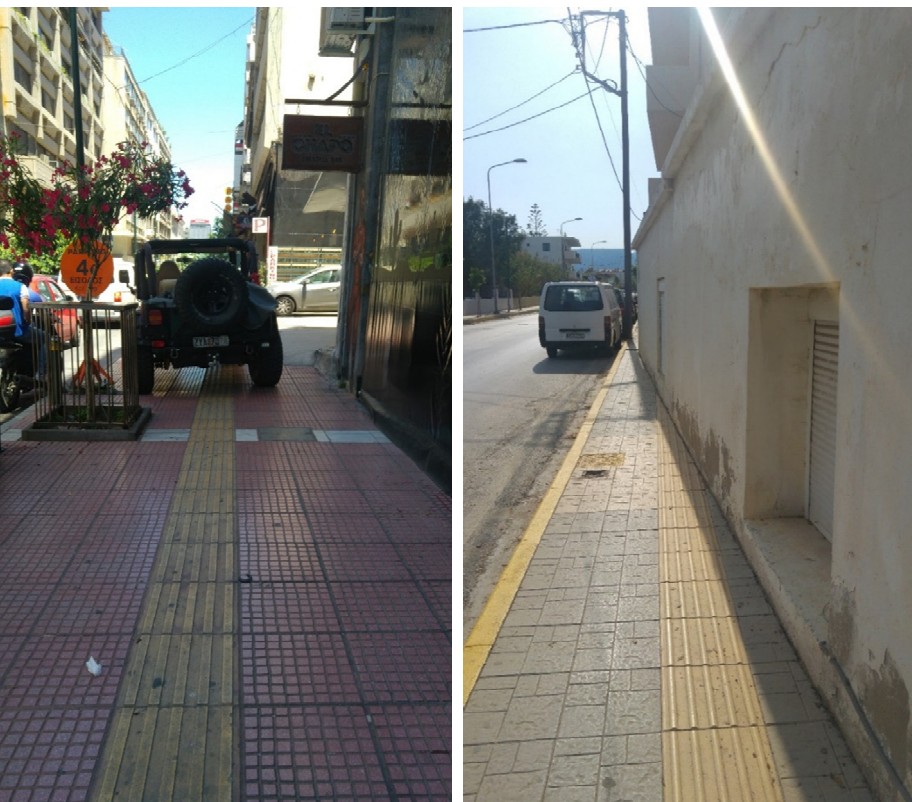

**Figure 11.** Obstacles difficult to analyze.

## 6. Impulse Noise Filtering

The proposed outdoor blind navigation system consists of two completely separated devices: (1) A wearable device and (2) an Android mobile application (an iOS version will follow). These subsystems communicate via a paired Bluetooth socket. The wearable device is responsible to collect the real-time field measurements and share them with the Android application in JSON format. As already mentioned, this data includes the user location according to the GPS/GLONASS system, as well as the sonar scan measurements. The Android application receives the data and transforms them into local model data, and navigation flow begins. An important achievement of the proposed system described in this work is to recognize and manage the way the blind person can avoid an obstacle along the walk route. This is realized through efficient sonar data analysis. Sonar measurements can occasionally present impulse noise due to field reflection or sensor controller transmission mistakes. The proposed system implements the well-known Median and Kalman filtering mechanisms to effectively remove such noise and enhance the reliability of the application. Such noise reduction is a typical pre-processing step to improve the results of later processing of the obstacle recognition algorithm. This section presents the details and performance of the implementation of the Median and the more computationally intensive Kalman filter on the target platform.

The main concept of the Median filter is to run through the measurements array entry by entry, replacing each entry with the median of neighboring entries. The pattern

of neighbors is called the "window", which slides, entry by entry, over the entire set of measurements. For one-dimensional (1D) signals, the most obvious window is just the first few preceding and following entries [29]. The Median filter demonstrates a regular O(n) complexity, including a mild sorting multiplicative constant factor depending on the sliding window size. For window size 5 the complexity of the algorithm becomes 5log5·n (see Figure 12).

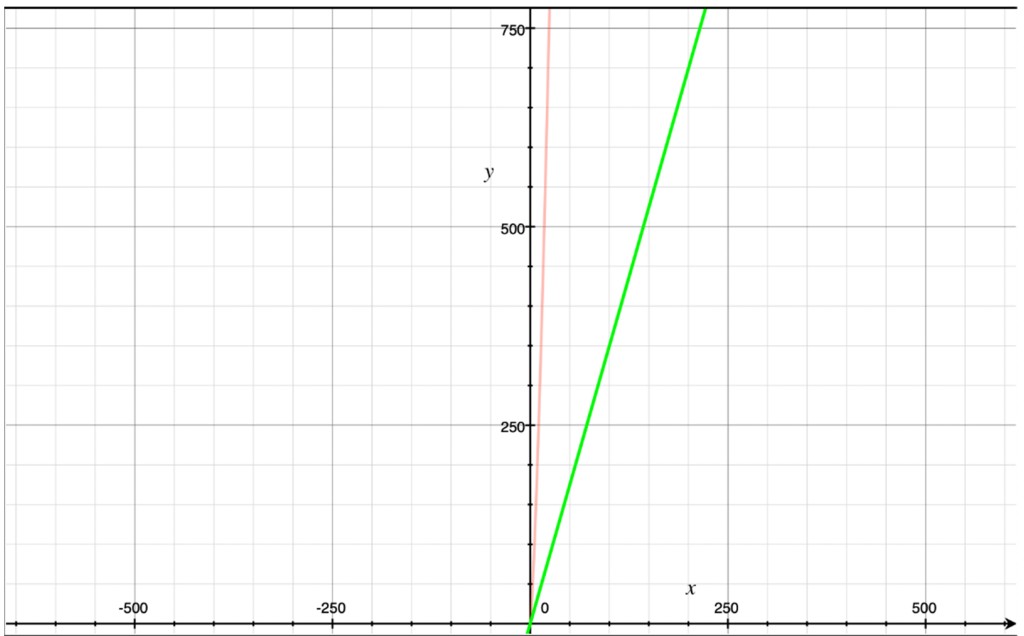

**Figure 12.** Complexity of Median and Kalman filter implementation.

The Kalman filter [30] is a statistic algorithm that likewise uses a series of measurements as input. It is based on the continued observation of a dynamic system over time, and it uses previous measurements to predict and correct inaccuracies at variables that tend to be more accurate than those based on a single measurement alone. It is a commonly used filter in dynamic systems either because measurements contain errors or are occasionally not accessible. The Kalman preprocessing in our obstacle recognition algorithm transforms field sonar measurements into square matrices which are input to the Kalman filter. The filter is applied in the same way in every new measurements matrix and the resulting "corrected" matrix is sequentially used as previous measurements matrix on next input. The critical issue in the Kalman filter implementation is the matrix creation and calculations (multiplication, transposition, etc.). The Kalman filter complexity in our implementation, due to many matrices (e.g., gain_matrix, noise_matrix, previous_matrix) with different sizes, is y = 5·×·$\sqrt{}$x+7x.

For example, *n* = 36 items MeasureArray creates a matrix size 6 × 6; therefore, looping the 36 items in both Median and Kalman filters has O(n$\sqrt{}$n) complexity. Figure 12 illustrates the complexity comparison of the two filters. As expected, the Median filter is less computationally intensive and is the favorite selection for sonar impulse noise removal extendedly preserving CPU power and battery life between charges on restricted mobile devices executing the sonar obstacle recognition algorithm. At the same time, as it will be obvious by the ensuing analysis, the Kalman filter can be effectively executed on the same target platform.

Following the brief theoretical analysis of the Median and Kalman filters, we tested them with real measurements. Wearable device configuration was modified to send data every second to the mobile host. This way we manage to gain more interrupts at the Android application. We performed the experiment for measurements size 4 and 36 in order to confirm our theoretical analysis. In our experiments, we used the Android Studio

Profiler [31] to evaluate the CPU and memory load effected by both filters using a different number of samples, as well as the pCloudy Mobile Application Performance Monitoring tool [32] to monitor the respective smartphone battery consumptions. We used a Nokia 8.1 Android One smartphone, running Android version 9, which is powered by a 3500 mAh battery.

Figures 13 and 14 illustrate through the Android Studio Profiler the hardware resource usage involving a four measurement input (2 × 2 Kalman matrix). We notice many process interrupts, including many UI interrupts in the main thread because our application further implements a google maps fragment at the UI, in case of assistive navigation from a sighted volunteer. The two filters look very similar as per CPU usage; however, evidently the Kalman filter should handle a higher process interrupt rate.

Figures 15 and 16 illustrate the hardware resource usage involving a 36 measurements input (6 × 6 Kalman matrix). CPU usage increase is easily noticeable, which implies more power consumption and memory usage. The growth of complexity and produced data is evident. This data should be analyzed in real-time by the blind navigation smartphone application.

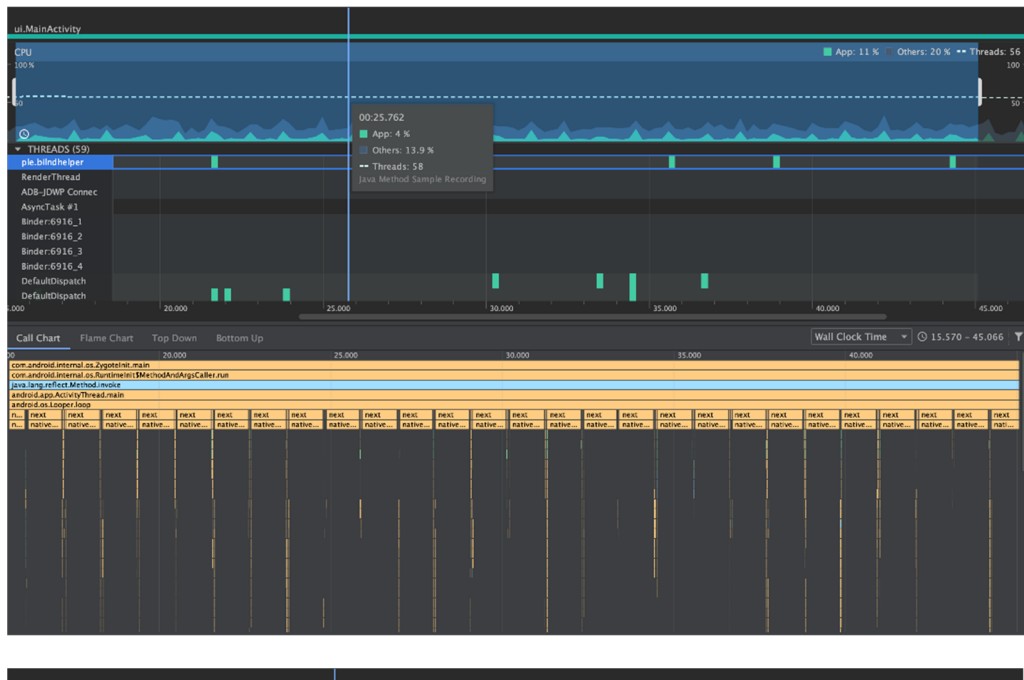

**Figure 13.** Median-4 filter CPU, memory and power load.

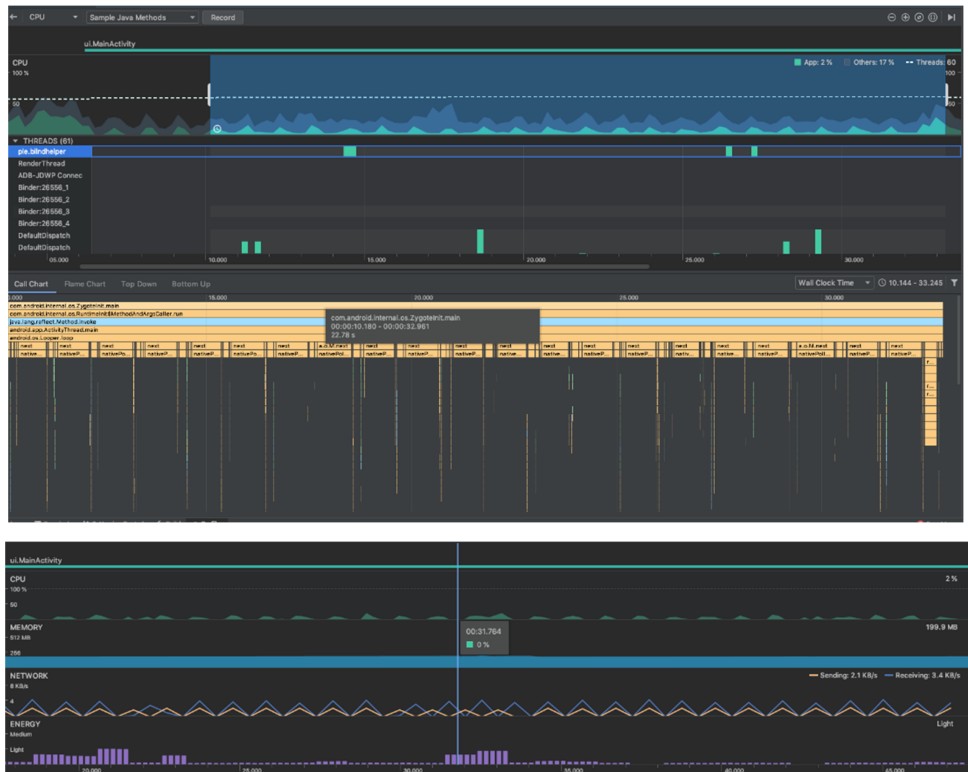

**Figure 14.** 2 × 2 Kalman filter CPU, memory and power load.

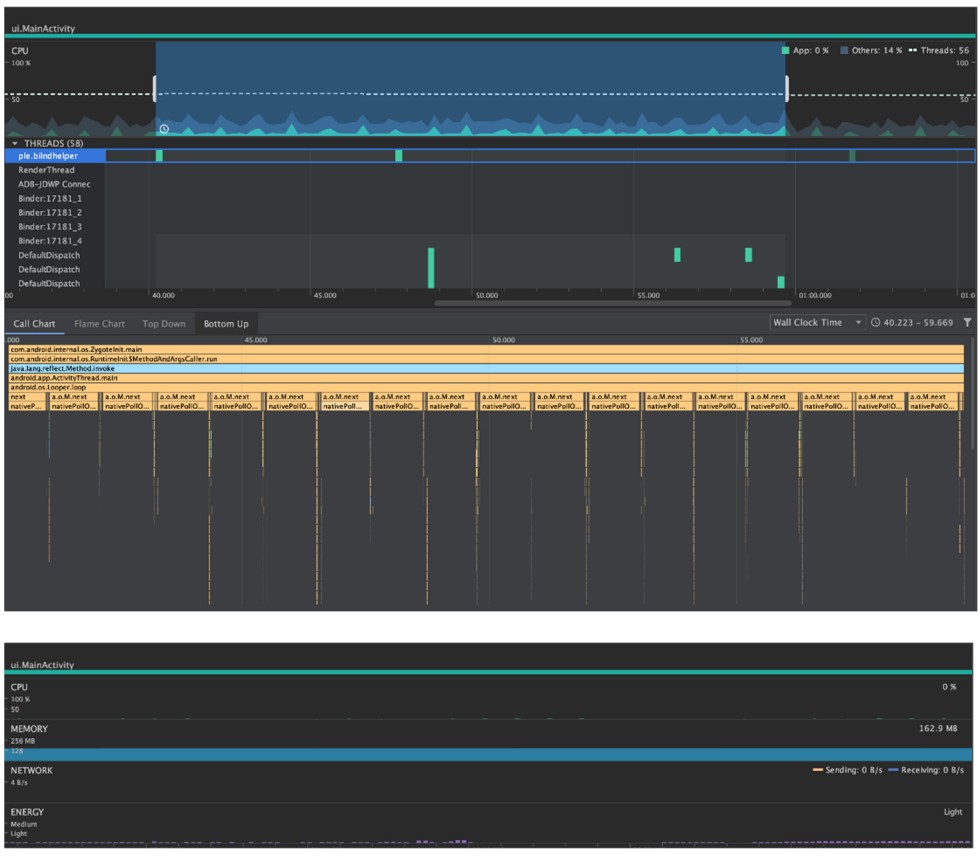

**Figure 15.** Median-36 filter CPU, memory and power load.

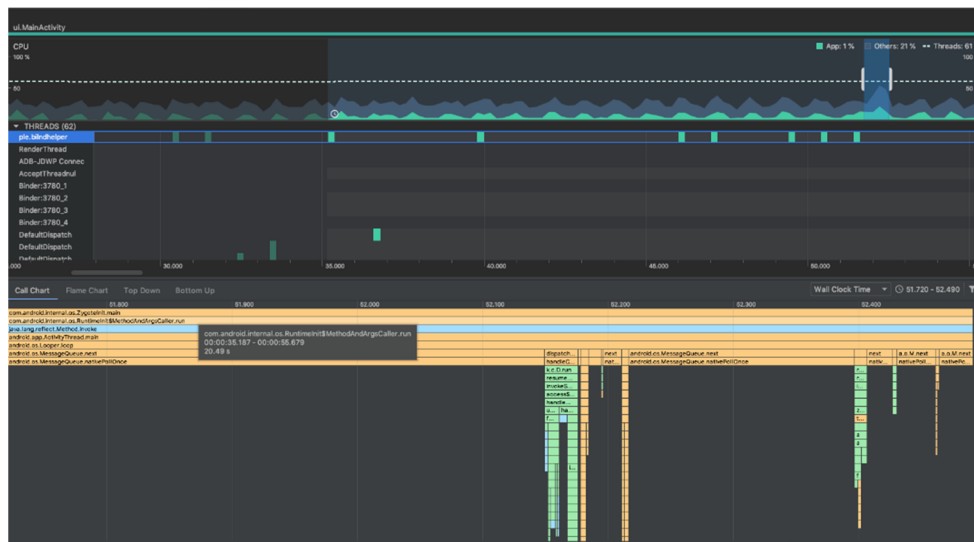

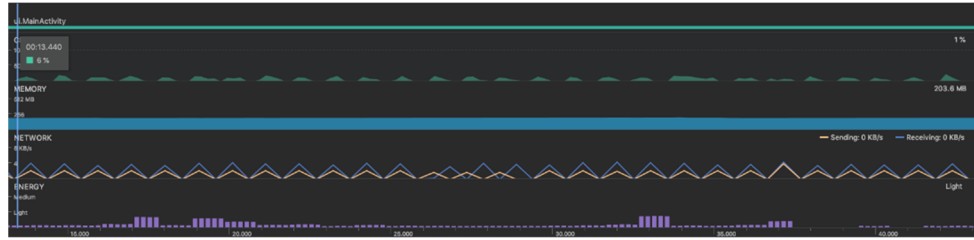

**Figure 16.** 6 × 6 Kalman filter CPU, memory and power load.

The Median filter demonstrates a linear complexity for both small and wide range windows. On the other hand, the Kalman Filter can be effectively executed on the proposed system even for wide range measurements; however, CPU load rises significantly compared to small measurements ranges. Kalman memory usage is always increased compared to Median due to the square matrices required for error correction. Furthermore, the Kalman filter relies on previous measures, while the Median filter is memoryless. Power efficiency is also an important factor since it should be reminded that the proposed systems is portable and powered by battery. Finally, the Kalman filter provides enhanced prediction abilities which can further combine different type of measurements input, such as from GPS, gyroscope, step metering sensors, etc., and it is important that it is supported efficiently in our implementation. Table 2 summarizes the CPU, memory and battery load of both filter implementations for different window sizes. Figures 17 and 18 illustrate the battery consumption in continuous time in percentage of the smartphone device (yellow line) and in mAh for the blind navigation application integrating the obstacle recognition algorithm with impulse noise filtering (red line), as well as in total for the smartphone device featuring a 3500 mAh battery (blue line). Battery life between charges was measured in the range between 59.2 h and 112.7 h.

**Table 2.** Summary of Median and Kalman filter footprint.

| Filter (Items) | CPU | Memory | Battery Consumption % (Device Related) |
|---|---|---|---|
| Median (4) | 2% | 118 MB | 2.46 |
| Kalman (4) | 5% | 196 MB | 3.19 |
| Median (36) | 5% | 162 MB | 2.98 |
| Kalman (36) | 12% | 210 MB | 6.11 |

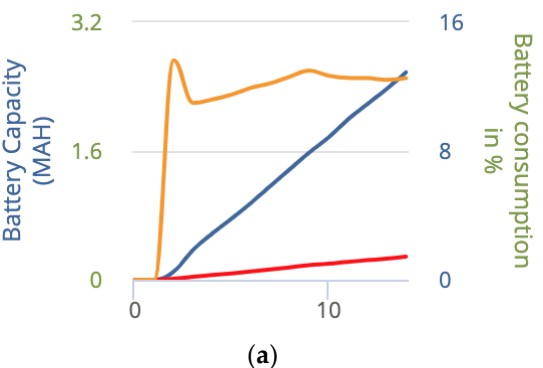 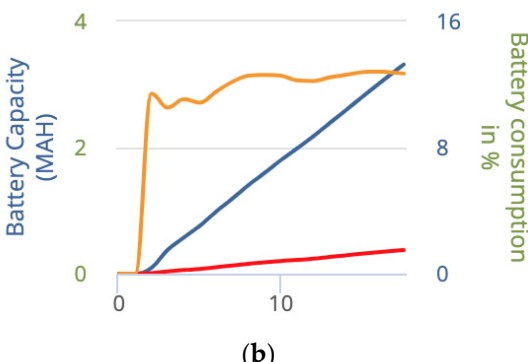

**(a)**      **(b)**

**Figure 17.** Battery consumption in % of device and mAh of (**a**) Median-4 and (**b**) Kalman-4 implementation (time axis in minutes).

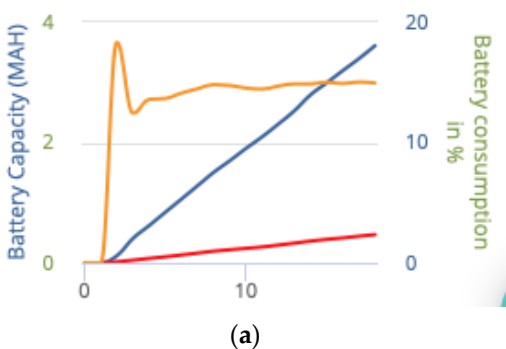 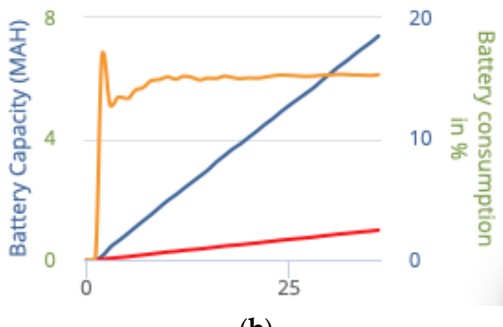

**(a)**      **(b)**

**Figure 18.** Battery consumption in % of device and mAh of (**a**) Median-36 and (**b**) Kalman-36 implementation (time axis in minutes).

The implemented Median and Kalman filters are pre-processing methods for data correction before the actual data analysis system process of the described algorithm. As this data will be used for obstacle tracking, any kind of impulse noise can dramatically deteriorate the recognition of obstacles and finally reduce the quality of service of the outdoor blind navigation application. This was occasionally evident in our experiments, e.g., E/Final Distance Kalman: 1476 (correct), while E/Final Distance No Filtering: 5155 (error) in one case involving impulse noise in the first sonar measurements. Such impulse noise typically includes the case when a sensor's signal incidents on a reflective surface and some portion of the signal are reflected back with a larger delay. Such false measurements can appear occasionally in the list among the correct sensor measurements and are removed through the sliding window of the applicable impulse noise filter. The environment surrounding noise cannot cause false measurements since the supersonic sensor operates in a different frequency band.

## 7. System Evaluation

This section presents the results of the evaluation of the proposed ultrasonic obstacle detection system using the median filter for the removal of the infrequent erroneous detections due to sensor noise. Several real scenarios are presented. In the following figure/image pairs, the figure on the left depicts the servo-motorized sensor measurements across a 60-degree range. The sensor performs a continuous fast scanning of the 60-degree view range in front of the blind person moving like a pendulum. The viewing angle can be set to 45 or 60 degrees, or to any other value in the system code. The figure on the left is the view of the debug activity of the sonar subsystem of the outdoor blind navigation app, which is not presented in the release mode of the app. The smartphone screen is split in two horizontal parts above and below the red line. Each part represents the raw (lower)

and median filtered (upper) detections of the sonar system. The vertical length of each part corresponds to a 4 m distance. The image on the right illustrates the corresponding real-world outdoor obstacle detection scenario. The blue line is composed of dashes of small width that each one corresponds to a ultrasonic detection at the specific rotation position of the servo-motor. The servo-motor rotates continuously the ultrasonic sensor left and right across the viewing range using a unitary rotation step of 2 degrees. Depending on the length between the user and the obstacle, the width of the detected obstacle and of the free space is geometrically calculated as explained in the previous sections.

Figure 19 illustrates the detection system response when a wall is ahead. Evidently, the median filter improves the system response, which is steady across the view range.

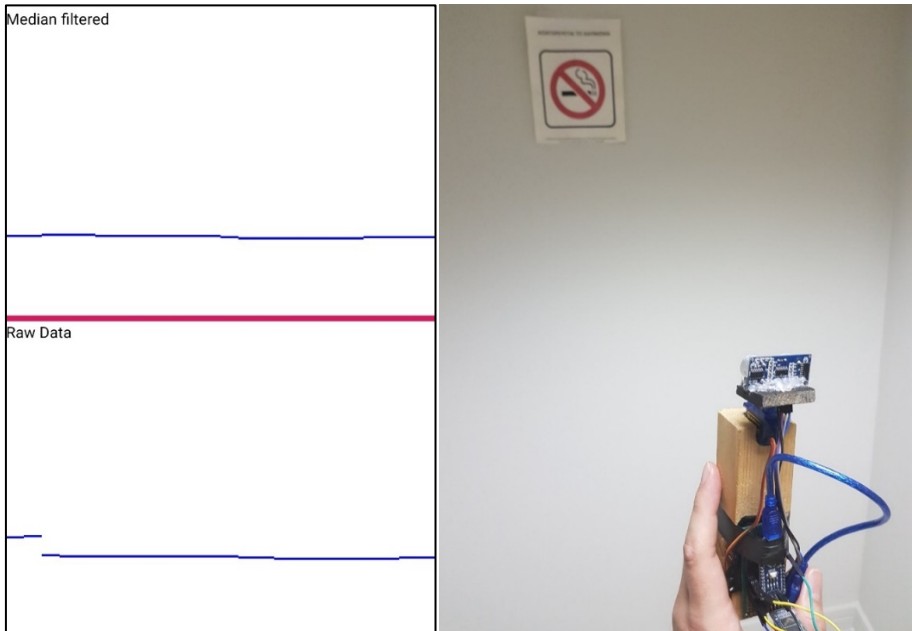

**Figure 19.** Measurements of the obstacle detection system facing a wall.

Figure 20 presents the obstacle detection system measurements when right ahead of the blind user, who is moving close to a wall along the pavement, there is a protrusion of a metallic cover of building constructions. The width of the obstacle in the right lateral view of the sonar is more than 30 cm and is detected 1.5 m in front of the user. The system detects the free space between the obstacle and the railings at the edge of the pavement, as indicated with the disruption of the blue line in the left and center part of the view area detection measurements. At the same time, the system detects the railings and parked cars at a larger distance.

Figure 21 illustrates the response of the sonar detection system when in front of the blind person, who is moving along the pavement, there is an erected scaffold for building restoration works. The sonar system detects the obstacle in the left and right lateral areas at 2.2 m distance in front of the user. The center area is clear, as is indicated by the absence of a corresponding blue line segment in the sonar scanner.

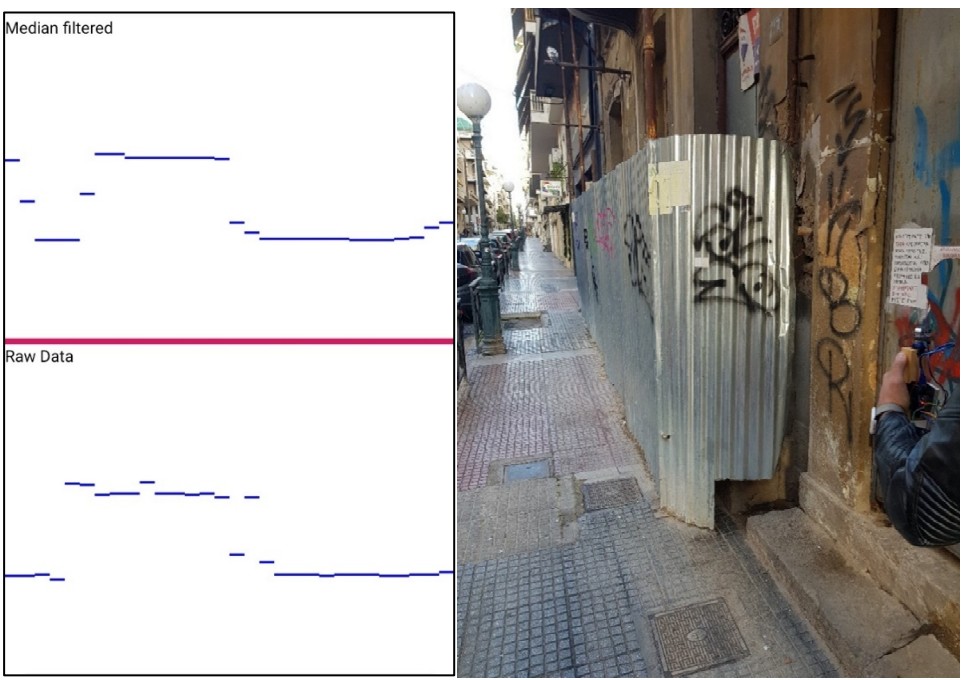

**Figure 20.** Response of sonar obstacle detection system.

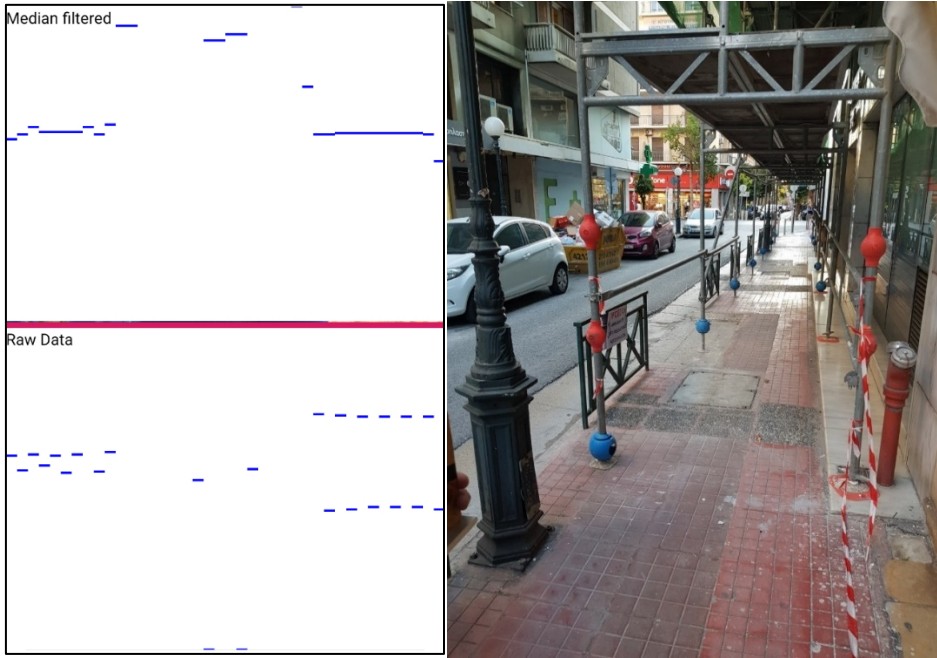

**Figure 21.** Response of sonar obstacle detection system.

Figure 22 illustrates the response of the sonar detection system in a similar situation as in the former case, in which the opening of the building scaffold in front of the blind person, who is moving along the pavement, is much narrower. The sonar device detects large width obstacles in the left front and in the right front area at a 1.2 m distance in front of the user, detecting a small opening in between, right ahead, through which the system will guide the user to pass through the scaffold pillars. The detected measurements in this case reveal a reliability issue of the sonar system integrating the selected cost-effective ultrasonic sensor, as the detected obstacle is considered much larger than it is. The reason

of this detection error is the inherent large detection angle of the ultrasonic sensor, which detects the small diameter pillar across a large rotation arc of the micro-servo-motor.

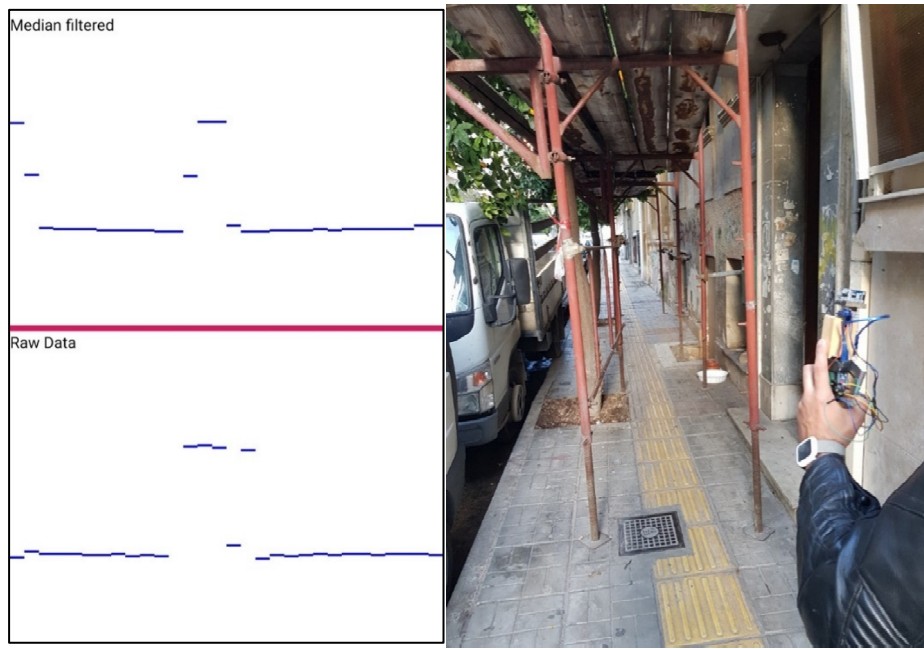

**Figure 22.** Response of sonar obstacle detection system.

Figure 23 illustrates the response of the sonar detection system when the blind user who is moving along the tactile route on the pavement meets a sign which is improperly located on the tactile route. Moreover, the route is interrupted by a construction barrier behind the sign and there is no tactile warning sign that can be sensed through the blind cane for bypassing the construction barrier. The sonar system detects the obstacles in front of the blind user, while the way out on the left, close to the edge of the pavement, although it is recognized by the sonar, does not look safe enough. Again, this use case reveals a reliability issue of the sonar system integrating the selected cost-effective ultrasonic sensor, which can be attributed to the limited ultrasonic directionality (wide beam) of the HC-SR04 sensor.

Figure 24 illustrates the response of the sonar detection system when the blind user attempts to enter a building whose entrance is surrounded by a construction works scaffold. The sonar system detects a large obstacle in the front right area and a small obstacle in the left lateral, while the clear central area detected in front of the blind user, indicated by the blue line interruption in the measurements graph, corresponds to a niche in the building wall which ends at the building entrance.

Figure 25 illustrates the response of the sonar detection system when it is scanning a clear area with a single human obstacle close to the blind user. The sonar system detects correctly the obstacle at a 1.5 m distance. The scarce detection measurements on the left are signaled by the parked car which is barely seen on the left of the picture. This particular use case validates that the sonar system integrating a wide beam cost-efficient ultrasonic sensor can detect reliably only sparse obstacles within a largely clear space in front of the user.

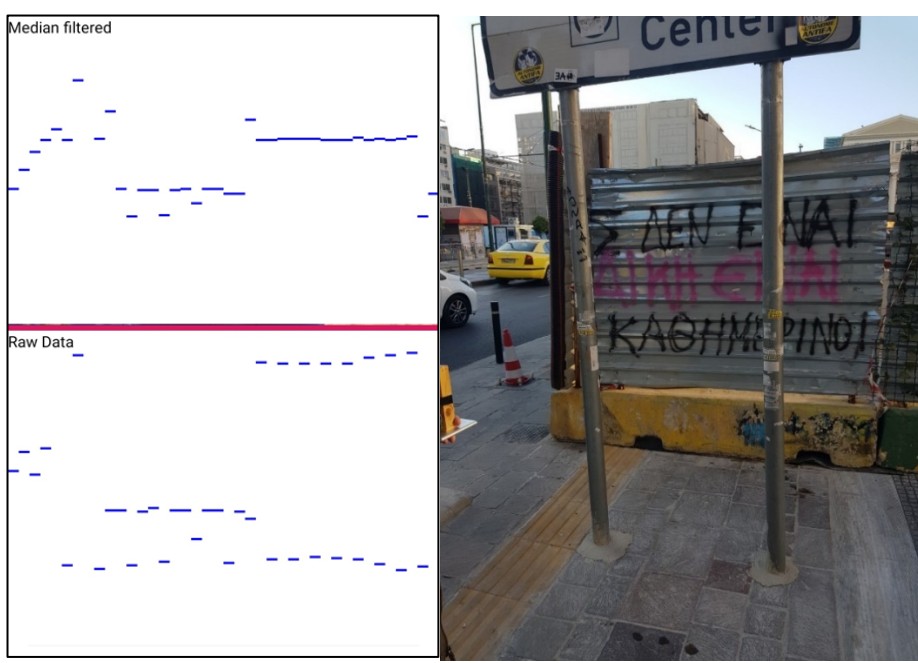

**Figure 23.** Response of sonar obstacle detection system.

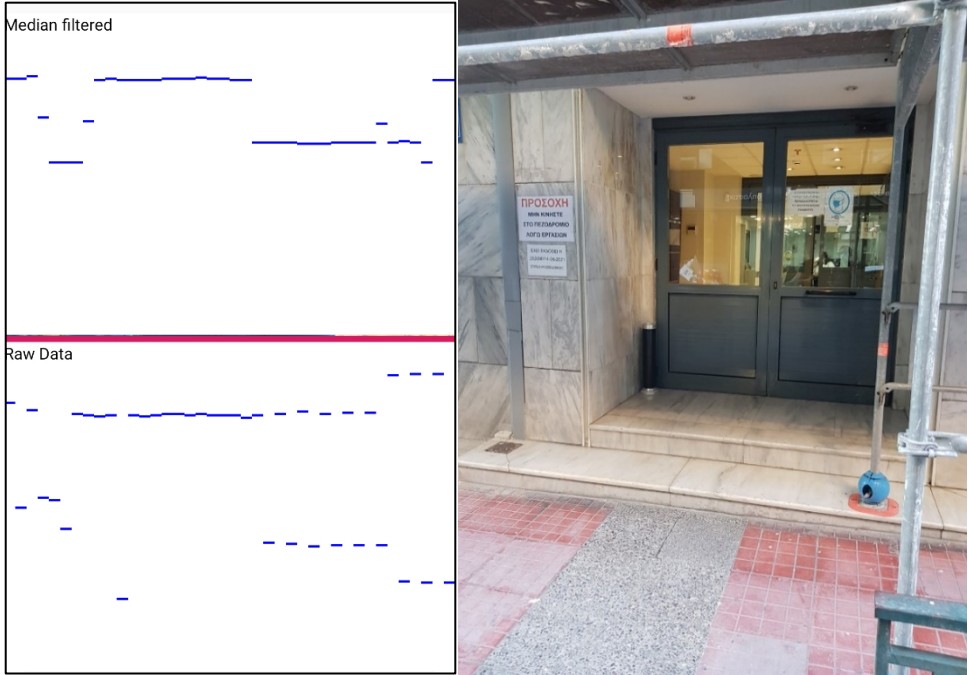

**Figure 24.** Response of sonar obstacle detection system.

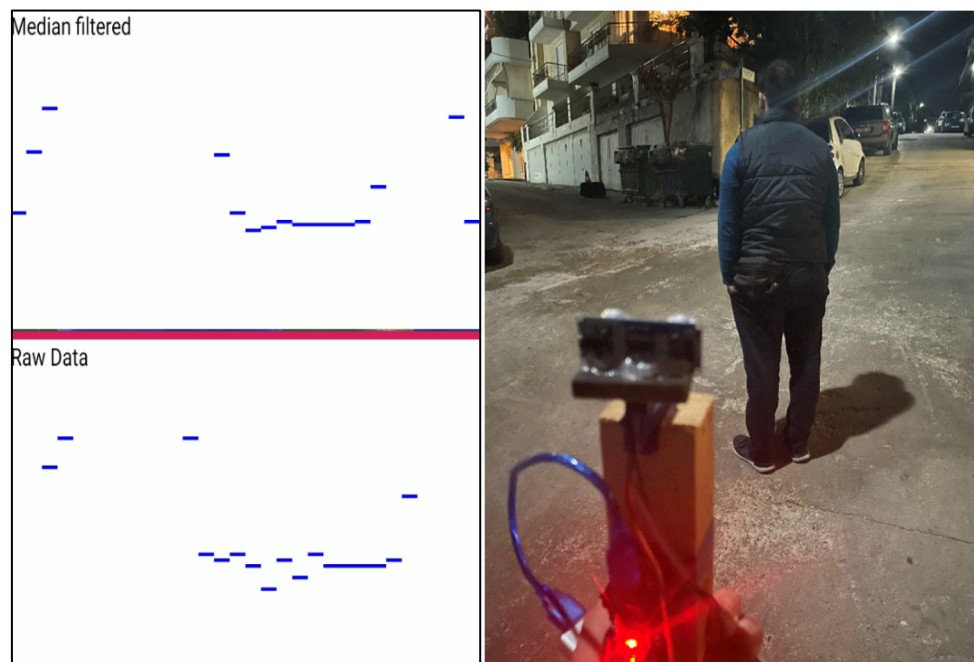

**Figure 25.** Response of sonar obstacle detection system.

It is noted that the presented trials were not repeated in different weather conditions. The speed of sound is slightly affected by air temperature, as it equals the square root of an almost constant factor affected by air molecular weight multiplied by temperature T in Kelvin degrees (sqrt($\gamma$RT)) [33]. For instance, the sound of speed is 343 m/s at 20 degrees of the Celsius scale, 337 m/s at 10 °C degrees C and 349 m/s at 30 °C. Distance calculation formula (1) in Section 4.1 will calculate a distance value of 2.094 m at 10 °C (2.023 m at 30 °C), instead of the correct distance of 2.058 m (a false ±35 mm deviation). Moreover, the impact of humidity in the air was not considered, as it is practically unimportant [34] and more difficult to test. The rule is that the more humidity in the air, the smaller its molecular weight will be, but again sound speed is slightly increased. Instead, the case of rain is an actual problem which may significantly deteriorate the performance of ultrasonic obstacle detection systems reporting shorter than the actual distances.

Last part of the evaluation is the power consumption of the sonar module. The MG90S micro-servo-motor current consumption is 550 mA at 5 V, while the working current of the HC-SR04 ultrasonic sensor is another 15 mA. The external microelectronic device of the outdoor blind navigation application developed in the MANTO project features either a 1240 mAh or a 3500 mAh battery in the typical configuration supporting solely the navigation functionality, excluding the sonar module. Stress battery discharging measurements were performed using the 3500 mAh battery in a demanding external device configuration sending constantly, every second, dense high precision GPS data to the smartphone application, addressing the Atmega microcontroller, the external GPS module and the Bluetooth module. The battery voltage level was measured every 2 min in room temperature conditions demonstrating an almost linear reduction from 4.3 V down to 3.67 V in 700 min. Charging of the external device will be necessary when the battery voltage level drops down to 3.5 V. When the sonar module is active, the sonar functionality alone drains the capacity of 3500 mAh in 6.2 h and its power consumption budget is added to the abovementioned basic external system power consumption. Evidently, the external microelectronic device can operate constantly for many hours before charging is required, even when integrating the sonar functionality, enhancing the user experience. Our measurements show that our external microelectronic device can demonstrate constant operation exceeding 12 h before recharging is required in the most demanding navigation scenario involving sending dense high precision GPS data to the smartphone application

every 1 s. Constant operation time with the sonar module active will exceed 4 h before charging is required, as, clearly, the servo-motor dominates the power budget of the external device. It is noted that a typical configuration setting for GPS reporting in our blind navigation application is 4 secs, which proves adequate considering the typical slow walking speed of a blind person. The external device configuration including the sonar module integrates the larger 3500 mAh battery.

## 8. Conclusions and Discussion

The above trials reveal that the ultrasonic sensor HC-SR04, which is widely used efficiently in robotic applications, is marginally efficient for robust obstacle detection in large distances up to the maximum sensor viewing distance. Its drawback is its low ultrasonic directionality. In practice, it is concluded that the proposed obstacle detection algorithm and micro-servo-motor sonar system performs very efficiently in an outdoor blind navigation framework in a dense city environment when integrating costly narrow/pencil beam ultrasonic sensors and setting correspondingly, i.e., optimized according to the sensor viewing angle and ultrasonic beam width, the rotation step of the servo-motor across a total viewing angle of 45 degrees of the sonar system along the user path. Such COTS sensors, mostly addressing industrial applications, can be found in the market at expensive prices. In outdoor blind navigation in a sparse city environment, along wide pavements, the proposed sonar system performance can be robust even when using cost-effective wider ultrasonic beam sensors.

The decisive factor regarding the reliability requirement of the proposed obstacle detection and avoidance mechanism is the width of the ultrasonic beam of the exploited sensor, rather than the detection distance. According to the datasheet, the cost-efficient HC-SR04 sensor can detect objects at a typical maximum distance of 4 m across a viewing angle of 15 degrees. At the same time, the datasheet depicts a wider ultrasonic beam diagram for the sensor, meaning that the sensor can detect objects inside the beam area, outside the stated viewing angle. More expensive, still affordable, COTS sensors, such as URM37, SRF08, SRF10, etc., can detect objects in distances up to 11 m with a similar viewing angle and beam width, although it is suggested to be configured to detect at a shorter distance for faster detection. The usual problem with such ultrasonic sensors is that despite their smaller viewing angle feature to assure detection at larger distances, nevertheless their ultrasonic beam widens at short distances yielding side detections outside the nominal viewing angle.

The proposed obstacle detection and avoidance guidance algorithm in the framework of outdoor blind navigation can demonstrate a very reliable detection performance exploiting long range narrow/pencil beam ultrasonic sensors. Such sensors are usually very expensive, such as the OMEGA ULR30 detecting objects at 10 m, having a viewing angle of 12 degrees, priced at \$700, as well as the BAUMER UNAM 70I6131/S14, featuring an extremely directional beam whose width is 40 cm (20 cm on either side of the central beam axis) and detection distance up to 6 m, priced at \$772. Furthermore, the ultrasonic sensors SICK UM30-214/215 which can detect objects at distances 5–8 m, feature a narrow beam of up to 1.6–2.4 m, priced at £456 and £483, respectively. A drawback of pencil beam ultrasonic sensors is that sometimes they fail to detect side obstacles that reflect the ultrasound at an angle beyond their detection capability.

The proposed system intends to address reliable guidance needs of the blind and visually impaired in public outdoor places using affordable standard modern technology provided by a typical low-end smartphone device. The article describes a state-of-the-art remote obstacle detection algorithm for a mobile application that will analyze the data received by an external cost effective low-power wireless microcontroller embedded device integrating an ultrasonic sensor module. Together the smartphone application and the external device will serve as a wearable that will help the navigation and guidance of blind and visually impaired people. A major objective, besides the offering by the overall system of a state-of-the-art navigation service to the blind and visually impaired, is to remotely

detect the existence and size of obstacles in the path of the user and to provide detailed reliable information through oral instructions about obstacles along the route of the visually impaired.

A further assessment of the presented system with additional sensors in real outdoor blind navigation conditions and use case scenarios is currently ongoing. The presented system is an integral part of the BlindRouteVision outdoor blind navigation application. The BlindRouteVision application was developed in the framework of the Greek R&D RCI1-00593 MANTO project (see the "Funding" section below for more details) [4]. It is concluded through this study and the developments in the MANTO project that a smartphone outdoor blind navigation application assisted by a wireless external embedded device can provide superior blind navigation performance against other well-known state-of-the-art GPS-based mobile blind navigation applications used by the blind and visually impaired community worldwide, such as BlindSquare and Lazarillo, which are both voicing apps paired with third-party navigation apps announcing detailed points of interest and intersections for reliable travel outside [35,36]. It can do so because of the enhanced centimeter-level GPS positioning and the inclusion of a modular component for the timely remote sensing of obstacles along the path of the blind and visually impaired.

The external wireless embedded device is designed in a modular way to allow the optional simple mounting of a servo-motorized ultrasonic sensor to provide remote obstacle sensing functionality. In that case, it is mandatory to wear the external device in a proper non-intrusive position for sensing, which is not required when not in use. The MANTO BlindRouteVision smartphone outdoor blind navigation application assisted by a small external embedded device aims to outperform widely appraised commercial applications in the most significant blind navigation aspects, such as precision, reliability and safety, and redefine the state-of-the-art in blind navigation.

**Author Contributions:** A.M. conceptualized and designed the proposed ultrasonic obstacle recognition algorithm and methodology, supervised the presented work and was the main author of the article and responsible for funding acquisition in the framework of the MANTO project (see Funding information in the next pararaph). J.L. implemented the proposed system software in the framework of his Erasmus engineering diploma final project. C.F. implemented the prototype device, the impulse noise filtering software, and integrated, extended, fine-tuned and validated the overall proposed system code on the target platform consisting of the smartphone and the external microprocessor device. He also created the system trials validation software. All authors have read and agreed to the published version of the manuscript.

**Funding:** This research was funded by the Greek RTDI State Aid Action RESEARCH-CREATE-INNOVATE of the National EPAnEK 2014–2020 Operational Programme "Competitiveness, Entrepreneurship and Innovation" in the framework of the MANTO project, under contract No. 593. The European Erasmus+ Student Mobility Framework funded the mobility of Jairo Llorente from the University of Valladolid, Spain, to the University of Piraeus, Greece, during the last year of his undergraduate engineering studies. The APC was waived by MDPI to contributors of the ACM PETRA 2019 International Conference on Pervasive Technologies Related to Assistive Environments.

**Institutional Review Board Statement:** Not applicable.

**Informed Consent Statement:** Not applicable.

**Data Availability Statement:** Not applicable.

**Acknowledgments:** The authors would like to specifically thank the Lighthouse for the Blind of Greece for their fruitful collaboration in the framework of the MANTO project. This work was partially supported by the University of Piraeus Research Centre.

**Conflicts of Interest:** The authors declare no conflict of interest. The funders had no role in the design of the study; in the collection, analyses, or interpretation of data; in the writing of the manuscript, or in the decision to publish the results.

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
