# Peer review of "Reliable Ultrasonic Obstacle Recognition for Outdoor Blind Navigation"

_technologies, doi:10.3390/technologies10030054_

Round 1

Reviewer 1 Report

The paper addresses an interesting and useful topic. It is more oriented to application than to research, but propose a good degree of novelty (even if it must be better explained, see later).

Enough content was added from the original conference version.

In my opinion, the paper has two main drawbacks that must be tackled before publication.

The first major drawback is that the paper does not present clearly the starting point of the proposed work. The paper is meant as an extension of a conference paper, but the latter is not cited. In the introduction, the authors cite the MANTO project, but they do not state whether it is a related work or the project in which the presented work has been developed. If the large part of the introduction is intended to present the related work, I suggest to split it into subsections and to add an appropriate title.

Moreover, the authors do not point out the limitations of the existing works; this would be useful to motivate the proposed approach.

The second major problem is about the conclusions. Section 7 seems to focus on the performance of the sensors (at least the half part of the section), a topic that has not be addressed in the paper and that is marginal to the core of the paper. Moreover, the discussion is confused: in line 722 the authors state that the used sensor “is marginally efficient for robust obstacle detection”, then in line 725 they say that it “performs very efficiently in an outdoor blind navigation framework”, and in line 732 they say that “the proposed sonar system performance can be robust”. The readers are confused.

If the authors aim at discussing the different sensors, an appropriate section must be provided before conclusions, and the discussion must be clear.

I have an additional, non critical, comment. In Section 4.3 the authors report the obstacles not detected or difficult to detect. My first question is: can the car sensors detect this kind of obstacles? I’m not sure, but I think my car can detect steps and stairs. Anyway, do the authors have some idea about how to solve the problem? Steps and stairs can be very dangerous for blind people.

Minor comments follow.

The abstract is very similar to the abstract of the conference version. I suggest to differentiate it to point out the extension of the journal version.

The beginning of the introduction addresses a very broad topic, the evolution of the circuits and chips. I would go directly to the topic of the paper, addressing the faced problem and not the possible solutions from the Moore law.

The sentence “the main use of the mobile phone was making calls to other contacts” (line34) is not clear. Other with respect to who?

In line 46 there is a ‘;’ while a ‘:’ is needed.

The sentence “Due to the parameters of the number of samples” (line 234) is not clear.

Subsections 3.2.1 and 3.2.2 have the same title.

In Section 3.2.2, what does “ensuing” in line 319 mean?

I suggest to use a different style for the names of the variables (e.g., kindOfMovement in line 379).

The format of the paper is quite ugly: almost blank pages must be avoided (for instance page 4), narrow images must be placed side by side and not one over the other (for instance, figure 2 and from 19 to 24). Some figures must be better centered (for instance, figure 5), in particular those that fall outside the page (for instance, figure 17).

The upper parts of figures from 19 to 24 must be bordered or framed, because sparse lines in the page are not easy to understand.

Reviewer 2 Report

Manuscript was very well conceived and prepared. Data are in real time applications with neat presentation.  

Reviewer 3 Report

  1. In your system an ultrasonic sensor HC-SR04 has been used which transmits and receives the sound waves. Since sound wave varies with environmental condition and your system has been designed for the outdoor activities, did you consider the environmental conditions such as cold and hot weather? If so are there any discrepancies of sensor readout accuracies?
  2. What happens if the sensor’s signal incidents on a reflective surface and some portion of the signal is reflected back with surrounding noise? Did you think about it during your experiment?
  3. you mentioned in section 3.1 that as the number of samples increases, the information becomes more reliable, yet the microcontroller requires more time. Is it possible to estimate the microcontroller’s power consumption in that case? Furthermore, you specified the power consumption of the smartphone, but there is no information on the power consumption of the microcontroller during different stages of operation. Is it possible to estimate? Could you make an comment on that.

Round 2

Reviewer 1 Report

The authors have addressed all my comments and the paper has been improved.